# *Chlamydia trachomatis*-containing vacuole serves as deubiquitination platform to stabilize Mcl-1 and to interfere with host defense

Annette Fischer[1], Kelly S Harrison[2], Yesid Ramirez[3], Daniela Auer[1], Suvagata Roy Chowdhury[1], Bhupesh K Prusty[1], Florian Sauer[3], Zoe Dimond[2], Caroline Kisker[3], P Scott Hefty[2], Thomas Rudel[1]*

[1]Department of Microbiology, Biocenter, University of Würzburg, Würzburg, Germany; [2]Department of Molecular Biosciences, University of Kansas, lawrence, United States; [3]Rudolf Virchow Center for Experimental Biomedicine, University of Würzburg, Würzburg, Germany

**Abstract** Obligate intracellular *Chlamydia trachomatis* replicate in a membrane-bound vacuole called inclusion, which serves as a signaling interface with the host cell. Here, we show that the chlamydial deubiquitinating enzyme (Cdu) 1 localizes in the inclusion membrane and faces the cytosol with the active deubiquitinating enzyme domain. The structure of this domain revealed high similarity to mammalian deubiquitinases with a unique α-helix close to the substrate-binding pocket. We identified the apoptosis regulator Mcl-1 as a target that interacts with Cdu1 and is stabilized by deubiquitination at the chlamydial inclusion. A chlamydial transposon insertion mutant in the Cdu1-encoding gene exhibited increased Mcl-1 and inclusion ubiquitination and reduced Mcl-1 stabilization. Additionally, inactivation of Cdu1 led to increased sensitivity of *C. trachomatis* for IFNγ and impaired infection in mice. Thus, the chlamydial inclusion serves as an enriched site for a deubiquitinating activity exerting a function in selective stabilization of host proteins and protection from host defense.

*For correspondence: thomas. rudel@biozentrum.uni-wuerzburg. de

**Competing interests:** The authors declare that no competing interests exist.

## Introduction

*Chlamydia trachomatis* is an obligate intracellular, gram-negative human pathogen that infects hundreds of millions of people every year to cause trachoma and blindness or sexually transmitted diseases (STD), which can lead to ectopic pregnancy and infertility. *Chlamydia* has a biphasic life-cycle starting with the infection of the host cell by the non-replicating elementary body (EB). Once the EB entered the cell, it differentiates into the metabolic active reticulate body (RB) and replicates inside a vacuole, the so-called chlamydial inclusion. To complete the developmental cycle, the RBs re-differentiate into the infectious EBs, which are released by host cell lysis or extrusion. During replication, the bacteria secrete several effector proteins through type two- (T2SS) or three-secretion systems (T3SS) to intrude in cellular processes for their benefit (*Subtil et al., 2005*; *Valdivia, 2008*). Proteins secreted via the T3SS may reside in the inclusion membrane (so-called Inc proteins) or other cellular compartments. Inc proteins in general interact with numerous different host proteins thereby orchestrating processes like vesicular trafficking and signaling cascades essential for inclusion formation and bacterial replication (*Elwell et al., 2016*). Due to the full dependence on the host cell for replication, *Chlamydia* established strategies to prevent apoptosis to sustain their favored environment (*Sharma and Rudel, 2009*). During bacterial entry into the host cell, the Raf/MEK/ERK and

PI3K/AKT survival signaling pathways are activated that in turn mediate the transcriptional up-regulation and stabilization of the anti-apoptotic Bcl-2 family member Mcl-1 (*Rajalingam et al., 2008*). In *C. trachomatis*-mediated apoptosis resistance in human cells, Mcl-1 is crucial because depletion of Mcl-1 sensitizes infected cells to cell death (*Rajalingam et al., 2008*; *Sharma et al., 2011*; *Rödel et al., 2012*).

Many cellular processes, including apoptosis, are orchestrated by the regulated stabilization and destabilization of proteins. For example, in order to induce apoptosis, various members of the anti-apoptotic Bcl-2 protein family have to be degraded to alter the ratio of pro- and anti-apoptotic proteins. A major pathway for regulating protein stability is the ubiquitin-proteasome system (UPS) that recognizes proteins modified with ubiquitin, an evolutionary conserved protein. Attachment of ubiquitin is catalyzed by a highly conserved enzymatic cascade in which the E3 ligase mediates substrate specificity. The most prominent K48-linked poly-ubiquitination is a signal for proteasomal degradation, whereas other forms of ubiquitination alter protein function or localization (*Vucic et al., 2011*). Ubiquitination is a strictly regulated but reversible process catalyzed by deubiquitinating enzymes (DUBs), which can be categorized in five classes and are either cysteine proteases or metalloproteases (*Nijman et al., 2005*; *Ha and Kim, 2008*).

The ratio of pro- and anti-apoptotic proteins of the Bcl-2 protein family regulating mitochondrial apoptosis signaling is adjusted by the UPS, too. Generally, the anti-apoptotic feature of Mcl-1 is based on its abundance in the cell and its degradation is an important step in the switch from survival to apoptosis. Here, the ubiquitin ligases MULE and the SCF$^{Fbw7}$ complex as well as the deubiquitinating enzyme USP9X mainly control the protein turnover of Mcl-1 (*Zhong et al., 2005*; *Ding et al., 2007*; *Inuzuka et al., 2011*; *Schwickart et al., 2010*). However, it is so far unknown by which mechanism *C. trachomatis* stabilize Mcl-1 upon infection.

Due to the fact that the UPS is such a powerful instrument to regulate cellular processes like the cell cycle, immune response or apoptosis, pathogens evolved several strategies to manipulate the UPS for their benefits. Many pathogens encode for effectors hijacking cellular components of the UPS or that possess ubiquitin ligase or deubiquitinase activity to interfere with ubiquitination or deubiquitination processes of cellular proteins (*Edelmann and Kessler, 2008*; *Ribet and Cossart, 2010*; *Furtado et al., 2013*). The chlamydial effector proteins ChlaDUB1 (CTL0247) and ChlaDUB2 (CTL0246), in the following termed Cdu1 and Cdu2, were demonstrated to have in vitro deubiquitinating and deneddylating activity (*Misaghi et al., 2006*; *Pruneda et al., 2016*). One of the Cdu1 targets was suggested to be IκBα, which may prevent NFκB signaling and immune response to ensure survival inside the host cell (*Le Negrate et al., 2008*).

In this study, Cdu1 enzyme was shown to localize to the inclusion membrane and stabilize Mcl-1 by deubiquitination. However, Mcl-1 protein levels decline only marginally in cells infected with a Cdu1 mutant strain of *Chlamydia* suggesting that keeping high Mcl-1 levels is regulated at multiple levels. Interestingly, the Cdu1 mutant was severely affected in its development upon IFNγ challenge in vitro and survival in mouse genital infections.

## Results

### Mcl-1 stabilization during chlamydial infection is related to altered ubiquitination

The anti-apoptotic Bcl-2 family member Mcl-1 has been identified in an RNA interference screen as a host cell protein important for apoptosis resistance in *Chlamydia*-infected cells (*Sharma et al., 2011*). Mcl-1 protein levels strongly increase upon 16 hr post infection (hpi) and remain high until the end of the chlamydial developmental cycle. Augmented Mcl-1 levels are the result of increased protein expression initially triggered by the Raf/MEK/ERK and the PI3K/AKT signaling pathway (*Rajalingam et al., 2008*). However, preventing Mcl-1 de novo protein synthesis by treatment of the cells with cycloheximide (CHX) or decreasing MCL1 gene transcription by blocking MEK/ERK signaling with the inhibitor UO126 revealed extended stabilization of Mcl-1 in infected cells (*Figure 1—figure supplement 1A–B*). The mechanism of prolonged Mcl-1 stabilization during the replicative stages of *C. trachomatis* infection is unknown.

To investigate the mechanisms of Mcl-1 stabilization during *Chlamydia* infection, protein ubiquitination patterns were examined during *C. trachomatis* infection. HeLa cells were infected with *C.*

*trachomatis* at a MOI of 1 and the proteasome inhibitor MG132 was added 20 hpi to avoid degradation of ubiquitinated proteins. One hour later, an apoptosis stimulus was given using TNFα/CHX and the cell lysate or precipitated Mcl-1 was analyzed for ubiquitination by immunoblotting. Compared to non-stimulated control cells, a mild increase in ubiquitination as a result of apoptosis signaling was detected after TNFα/CHX treatment in uninfected, but not in infected cells (*Figure 1A and C*). Interestingly, Mcl-1 ubiquitination was reduced in infected cells independently of the TNFα/CHX treatment (*Figure 1B,C* and *Figure 1—figure supplement 2A*). Reduced ubiquitination protects Mcl-1 from proteasomal degradation and may lead to Mcl-1 accumulation, as previously observed (*Rajalingam et al., 2008*). Since even the mild increase in Mcl-1 levels causes protection from apoptosis induction (*Figure 1—figure supplement 2B*), this stabilized Mcl-1 will interfere with apoptosis induction in infected cells. The anti-apoptotic function of Mcl-1 depends on its interaction with pro-apoptotic BH3-only proteins to block apoptosis induction (*Kim et al., 2006*). Bim is known to interact with Mcl-1 and both could be co-immunoprecipitated from cells infected with *C. trachomatis* (*Figure 1—figure supplement 2C*) suggesting that Mcl-1 is active in sequestering Bim in infected cells. Many cancer cells have increased levels of Mcl-1 protein due to the misregulation of cellular UPS factors like MULE, USP9X and components of the SCF$^{Fbw7}$ complex (*Schwickart et al., 2010*; *Zhong et al., 2005*; *Inuzuka et al., 2011*) which promotes apoptosis resistance (*Zhang et al., 2002*; *Adams and Cooper, 2007*). However, all these ubiquitin-modifying proteins were present at similar levels in control and *C. trachomatis*-infected HeLa cells (*Figure 1—figure supplement 2D–E*). The ubiquitin ligase activity of SCF E3 ligase complexes can be altered by neddylation of the Cullin subunit, but no difference in the protein level of Cullin 1 or its modification by Nedd8 was observed in *C. trachomatis*-infected cells compared to uninfected control cells.

Since the main regulators of Mcl-1 ubiquitination had been ruled out, we hypothesized that the chlamydial deubiquitinating enzyme Cdu1 (*Misaghi et al., 2006*) is playing a role in stabilization of Mcl-1. Cdu1 is known to be expressed 16 hpi (*Le Negrate et al., 2008*) when Mcl-1 levels start increasing in infected cells. Time course experiments confirmed a simultaneous Mcl-1 stabilization and Cdu1 expression (*Figure 1D*), indicating that Cdu1 might be responsible for Mcl-1 stabilization by deubiquitination and protection from proteasomal degradation.

## Structure of Cdu1

Sequence alignments clearly allocate Cdu1 into the family C48 of the CE clan of cysteine proteases. The peptidases share a common architecture toward the catalytically active C-terminus, while the N-terminus is weakly conserved (*Mossessova and Lima, 2000*) and members include the Ulp1 protease, from *Saccharomyces cerevisiae* (*Li and Hochstrasser, 1999*) as well as the Sentrin-specific proteases (SENPs) from higher eukaryotes (*Yeh et al., 2000*).

*C. trachomatis* Cdu1 is a 401 amino acid protein with a single-pass transmembrane domain located towards the N-terminus and a predicted flexible area up to residue 167. In situ proteolysis and LCMS analysis identified a stable Cdu1 construct that displayed full proteolytic activity (*Figure 2—figure supplement 1A*) to pursue its structural characterization. The Cdu1 (155-401) fragment crystallized in space group P2$_1$ (*Supplementary file 1*) with two molecules in the asymmetric unit and at a resolution of 1.7 Å. Each monomer is composed of a central four-stranded β-sheet, which is surrounded by eight α-helixes (*Figure 2A*) resembling the architecture of the C48 proteases. The catalytic cysteine, Cys345, is located at the N-terminus of α-helix F, which is 'sandwiched' in between a set of three anti-parallel β-strands (1,2,3) and two extra α-helices, A and G, which are perpendicular to each other. β-strands 2 and 3 also contain the two other residues of the catalytic triad, His275 and Asp292. Very recently, the structure of Cdu1 was also solved by Pruneda et al. at a resolution of 2.1 Å (*Pruneda et al., 2016*). Their fragment consists of residues 130 to 401 and can be superimposed with our Cdu1 structure with an rms deviation of 0.563 Å including residues 161 to 401 using the program LSQ superpose in COOT (*Kabsch, 1976*). Due to the difference in length of the construct the structure of Pruneda et al. contains an additional α-helix at the N-terminus which is not present in our Cdu1 structure (*Pruneda et al., 2016*).

A search for other related structures utilizing the program DALI (*Holm and Sander, 1993*) indicated that Cdu1 (155-401) shares a high degree of structural similarity with Ulp1 (*Mossessova and Lima, 2000*) despite the fact that they only share 17% sequence identity within a stretch of 221 amino acids. The crystal structure of Ulp1 in complex with the ubiquitin like protein SMT3 (*Figure 2B*) reveals the direction in which the C-terminal glycine of SMT3 approaches the active site

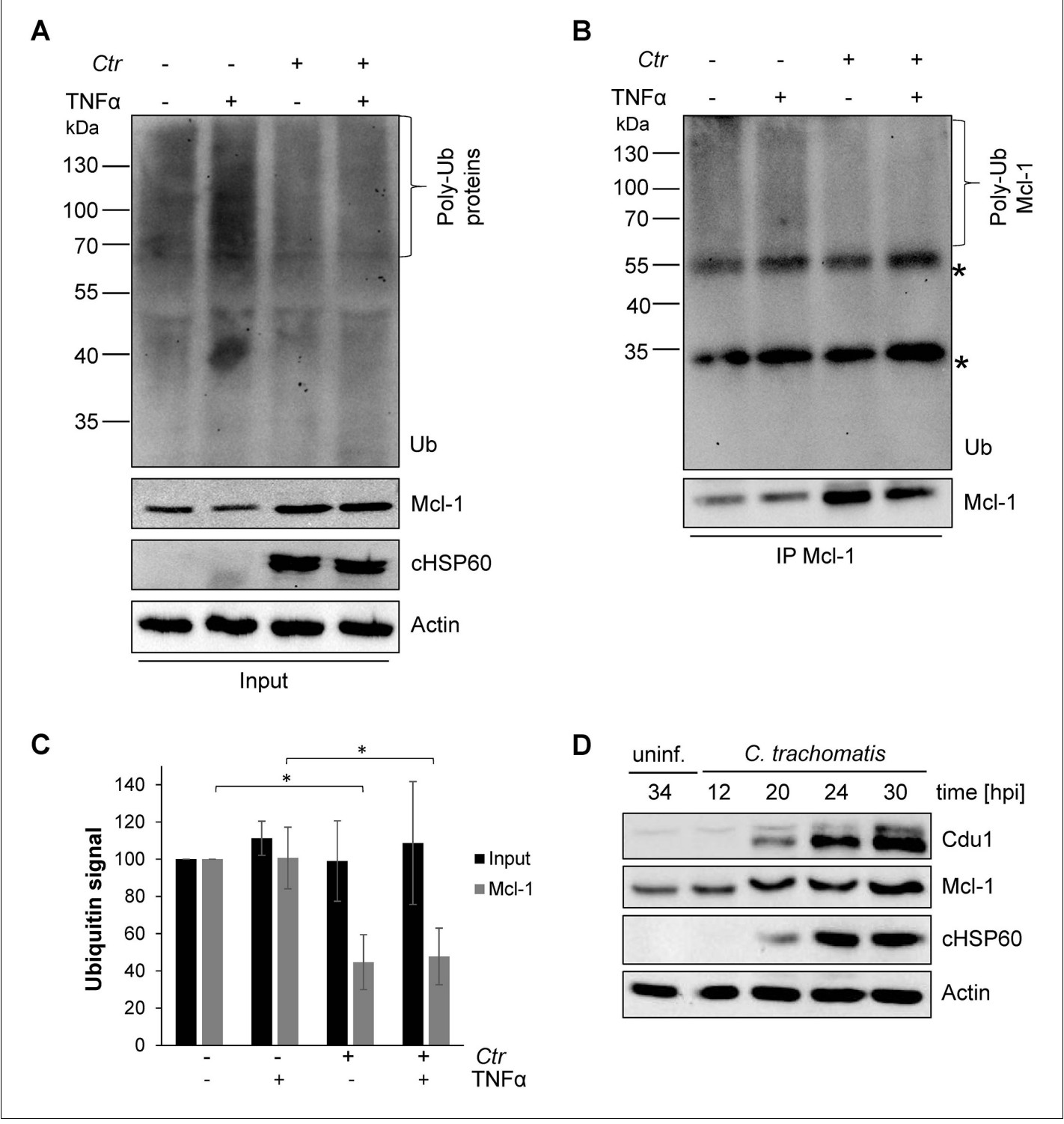

**Figure 1.** Mcl-1 is deubiquitinated during *C. trachomatis* infection. (**A**) HeLa cells were infected with *C. trachomatis* (*Ctr*) (MOI 1) and 20 hpi the proteasome inhibitor MG132 (20 μM) was added and apoptosis was induced by treatment with TNFα/CHX for 2.5 hr. Cells were lysed in RIPA-buffer under stringent conditions and the global ubiquitination status of proteins was tested by immunoblot analysis. Immunoblots for Actin, chlamydial HSP60 (cHSP60) and Mcl-1 function as input controls. Depicted are representative immunoblots. (**B**) Mcl-1 was immunoprecipitated under stringent conditions using a monoclonal anti-Mcl-1 antibody and the ubiquitination pattern was visualized by immunoblotting. Shown are representative immunoblots. The asterisks mark the signal from the heavy and light chains of the IgG antibody. (**C**) Quantification of ubiquitination of whole cell lysates or immunoprecipitated Mcl-1 (**A** and **B**). The graph shows mean values from three individual experiments ± SD. The significance was calculated with the
*Figure 1 continued on next page*

*Figure 1 continued*

student's T-test *p<0.05; **p<0.01. (D) Infection time course experiment with *C. trachomatis*-infected HeLa cells. Time of Cdu1 expression and Mcl-1 stabilization was analyzed by immunoblot. The second band visible in the Cdu1 blot may originate from a so far unknown post translational modification of Cdu1 or an unrelated protein. Decoration with chlamydial HSP60 and Actin antibodies serves as a control for infection and loading, respectively. See also *Figure 1—source data 1*.

The following source data and figure supplements are available for figure 1:

**Source data 1.** Raw data for quantitative analysis of Mcl-1 ubiquitination level shown in *Figure 1*.

**Figure supplement 1.** Mcl-1 is stabilized during chlamydial infection.

**Figure supplement 1—source data 1.** Raw data for quantitative analysis of Mcl-1 level in UO126-treated HeLa cells infected with *C. trachomatis* shown in *Figure 1—figure supplement 1*.

**Figure supplement 2.** Cellular components of the UPS play minor role in *Chlamydia*-mediated Mcl-1 stabilization.

cysteine. This path seems to be conserved between the two enzymes. Remarkably, however, a small loop between β-strands 1 and 2 (shown in yellow) in Ulp1 contrasts with the much bigger α-helix D in Cdu1 (*Figure 2A*) in between the equivalent β-strands of both structures. The same phenomenon can also be observed when comparing Cdu1 with the human SENP8 protein in complex with a Nedd8 aldehyde (*Reverter et al., 2005*), with whom Cdu1 shares 11% identity within 208 amino acids. Although larger than in Ulp1, the loop between β-strands 1 and 2 fails to adopt the helical architecture observed in Cdu1 (*Figure 2C*). In contrast to this difference, the residues within the active site are highly conserved (*Figure 2D and E*), further supporting the notion that Cdu1 forms a complex with ubiquitin in a similar way as observed between SENP8 and Nedd8 or Ulp1 and SMT3, respectively. The additional α-helix D in Cdu1, a feature not observed in the other structures, however, may play a regulatory role in substrate binding or recognition. This hypothesis is further strengthened by the analysis of Pruneda et al. where deletion of this region (residues 250 to 272) prevented Cdu1 from binding ubiquitin/Nedd8 suicide probes (*Pruneda et al., 2016*). Importantly, an insertion of identical length as observed in Cdu1 seems to be present in the Cdu1 proteins encoded by the genomes of *C. suis, C. muridarum, C. psittaci, C. gallinaceai* and Cdu2 from *C. trachomatis* suggesting that it is a conserved feature across the chlamydial deubiquitinases (*Figure 2—figure supplement 1B*).

## Cdu1 and Mcl-1 interact with each other

To investigate if Mcl-1 is a substrate of Cdu1 and whether these two proteins interact in infected cells, immunofluorescence confocal microscopy was performed to co-localize both proteins in *C. trachomatis*-infected cells. A rabbit polyclonal antiserum directed against recombinant Cdu1 detected Cdu1 inside the inclusion 24 hpi co-localizing with bacterial particles. Major amounts of Cdu1 were also detected at the surface of the inclusion co-localizing with the inclusion membrane protein IncA, where it might be anchored by its N-terminal transmembrane domain (*Figure 3A and B*). However, Cdu1 could not be detected in the host cell cytoplasm as it had been described before (*Le Negrate et al., 2008*). To obtain further proof for a localization of Cdu1 around the inclusion, recombinant *Chlamydia* were generated which express a FLAG-tagged version of Cdu1 that could be detected by an anti-FLAG antibody. Since Cdu1 is subjected to a tight transcriptional regulation during the chlamydial developmental cycle, the expression of the recombinant gene under the control of the original *cdu1* promoter was established. In a first approach, the plasmid pAH1 (*Figure 3—figure supplement 1A*) was designed, which cannot replicate in *Chlamydia* but can integrate in part into the genome by homologous recombination. The expected arrangement of the *cdu1* gene (*Figure 3C*) including the FLAG-tag at the C-terminus after recombination in the transformed new *Chlamydia* strain (*Ctr* Cdu1-FLAG) was verified by PCR and sequencing, as well as by Southern hybridization (*Figure 3—figure supplement 1B–D*). *Ctr* Cdu1-FLAG expressed only the FLAG-tagged version of Cdu1 (*Figure 3D* and *Figure 3—figure supplement 2A*), which was secreted to the surface of the inclusion (*Figure 3E*). The accessibility of the C-terminal catalytic domain of Cdu1

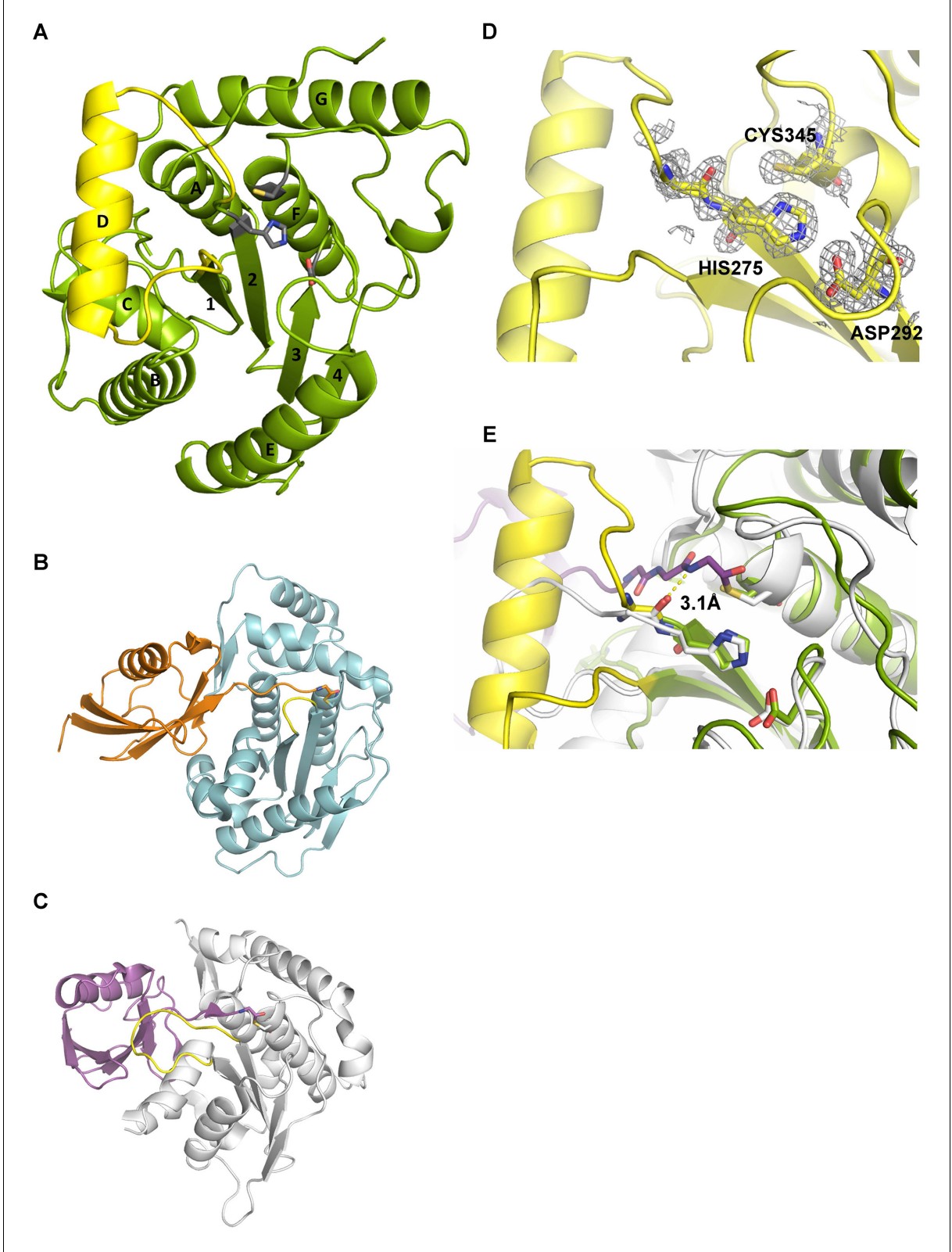

**Figure 2.** Structural features of Cdu1. (A) Overall structure of Cdu1 showing the catalytic triad (in stick representation) and the lid-helix motif between β-strands 1 and 2 (yellow). (B) Structure of the Ulp1 (cyan)-SMT3 (orange) complex (PDB 1EUV). The loop between β-strands 1 and 2 is shown in yellow. (C) Structure of the SENP8 (gray) – Nedd8 (purple) complex (PDB 1XT9). The loop between β-strands 1 and 2 is shown in yellow. (D) 2Fo – Fc omit maps

*Figure 2 continued on next page*

Figure 2 continued

for the catalytic triad of Cdu1 is contoured at an rmsd level of 1.0. The backbone carbonyl of Ser 274 next to the active site histidine points to the binding site of the terminal Gly of Ub/Nedd8. (E) Superimposition of the Cdu1 active site (green) with the SENP8-Nedd8 complex.

The following source data and figure supplements are available for figure 2:

**Figure supplement 1.** Enzymatic activity of Cdu1 used for crystallization and sequence alignment of the conserved protease domain.

**Figure supplement 1—source data 1.** Raw data for analysis of relative enzymatic activity of Cdu1 in Ub-AMC hydrolysis assay shown in *Figure 2—figure supplement 1*.

to the host cell cytosol was demonstrated by antibody transfection experiments (*Figure 3F*). Recombinant expression of Cdu1 in *Chlamydia* did not change its infectivity, as assessed by the efficiency and the duration of the developmental cycle, nor was the inhibition of host cell apoptosis affected (*Figure 3—figure supplement 2B–E*). In addition, the kinetics of Cdu1-FLAG expression and localization during the developmental cycle was not altered confirming the expression of the recombinant protein under the native promoter (*Figure 3—figure supplements 3* and *4*). Moreover, transcriptomics analysis of *C. trachomatis*-infected cells revealed that the insertion of the selection cassette at the *cdu1* locus should not affect the downstream gene *cdu2* since it is controlled by its own promoter (*Albrecht et al., 2010*).

Mcl-1 is known to be located in the cytoplasm and to a certain degree in a complex with BAK at intracellular membranes (*Thomas et al., 2010*). This localization was confirmed by immunofluorescence staining of uninfected cells. Interestingly, in *C. trachomatis*-infected cells high amounts of Mcl-1 were detected in close proximity to the inclusion (*Figure 4A,B* and *Figure 4—figure supplement 1A–C*) where Cdu1 is located (*Figure 3A and B*). This Mcl-1 Cdu1 co-localization was also detected in cells infected with *Ctr* Cdu1-FLAG (*Figure 4C*) or WT-infected cells expressing a GFP-Mcl-1 fusion protein (*Figure 4D*) ruling out unspecific binding of the antibody.

The interaction of both proteins was then analyzed by co-immunoprecipitation of endogenous Mcl-1 together with Cdu1-FLAG from HeLa cells infected with *Ctr* Cdu1-FLAG (*Figure 4E*). Mcl-1 but not p53, another high turnover protein constantly targeted for ubiquitination, precipitated together with Cdu1 confirming Cdu1 substrate specificity for Mcl-1. To rule out any non-specific binding of Mcl-1 to the FLAG-tag, HeLa cells were infected with *C. trachomatis* strains *Ctr* pTet/Cdu1-FLAG, *Ctr* pIncA-FLAG and *Ctr* pTet/Cdu2-FLAG overexpressing the indicated FLAG-tagged proteins. Endogenous Mcl-1 only co-precipitated together with Cdu1-FLAG but not with the other proteins from cells infected with the respective *Chlamydia* strain (*Figure 4F*). Further proof was provided by co-immunoprecipitation of overexpressed Cdu1-FLAG and endogenous Mcl-1 in uninfected HEK-293T cells (*Figure 4—figure supplement 1D*). The direct binding of both proteins to each other was addressed using recombinant purified proteins in in vitro binding assays. GST-tagged Mcl-1 was bound to glutathione-sepharose beads and incubated with equal amounts of His-tagged Cdu1 or control proteins (His-Cdu2 and His-GroEL). GST-Mcl-1 bound exclusively to His-Cdu1, demonstrating a direct interaction of these two proteins (*Figure 4G*).

## Cdu1 reduces ubiquitinated forms of Mcl-1 in cell culture

Since Mcl-1 and Cdu1 interact with each other and share the same subcellular localization during infection, Cdu1 expression should stabilize Mcl-1 even in the absence of infection. HEK-293T cells were transfected with Cdu1-FLAG or an inactive mutated form of the enzyme that carried an amino acid exchange of the catalytically active cysteine 345 to alanine (Cdu1(C345A)). Overexpression of Cdu1-FLAG led to a specific stabilization of Mcl-1 since Bcl-2, another anti-apoptotic Bcl-2 family member, and several inhibitor of apoptosis proteins (IAPs) were not affected by Cdu1-FLAG overexpression (*Figure 5A*). Overexpression of the mutant Cdu1(C345A) could not stabilize Mcl-1 to the same extent as the wild-type enzyme (*Figure 5A and B*), suggesting that chlamydial Cdu1 alone is able to stabilize Mcl-1, very likely by deubiquitination. To confirm Mcl-1 deubiquitination by Cdu1 expression, Mcl-1 was immunoprecipitated by its myc-tag under denaturing conditions from HEK-293T cells expressing HA-ubiquitin, myc-Mcl-1 and Cdu1-FLAG, Cdu1(C345A)-FLAG or empty vector control. The degree of poly-ubiquitination was higher if Mcl-1 was precipitated from cells co-

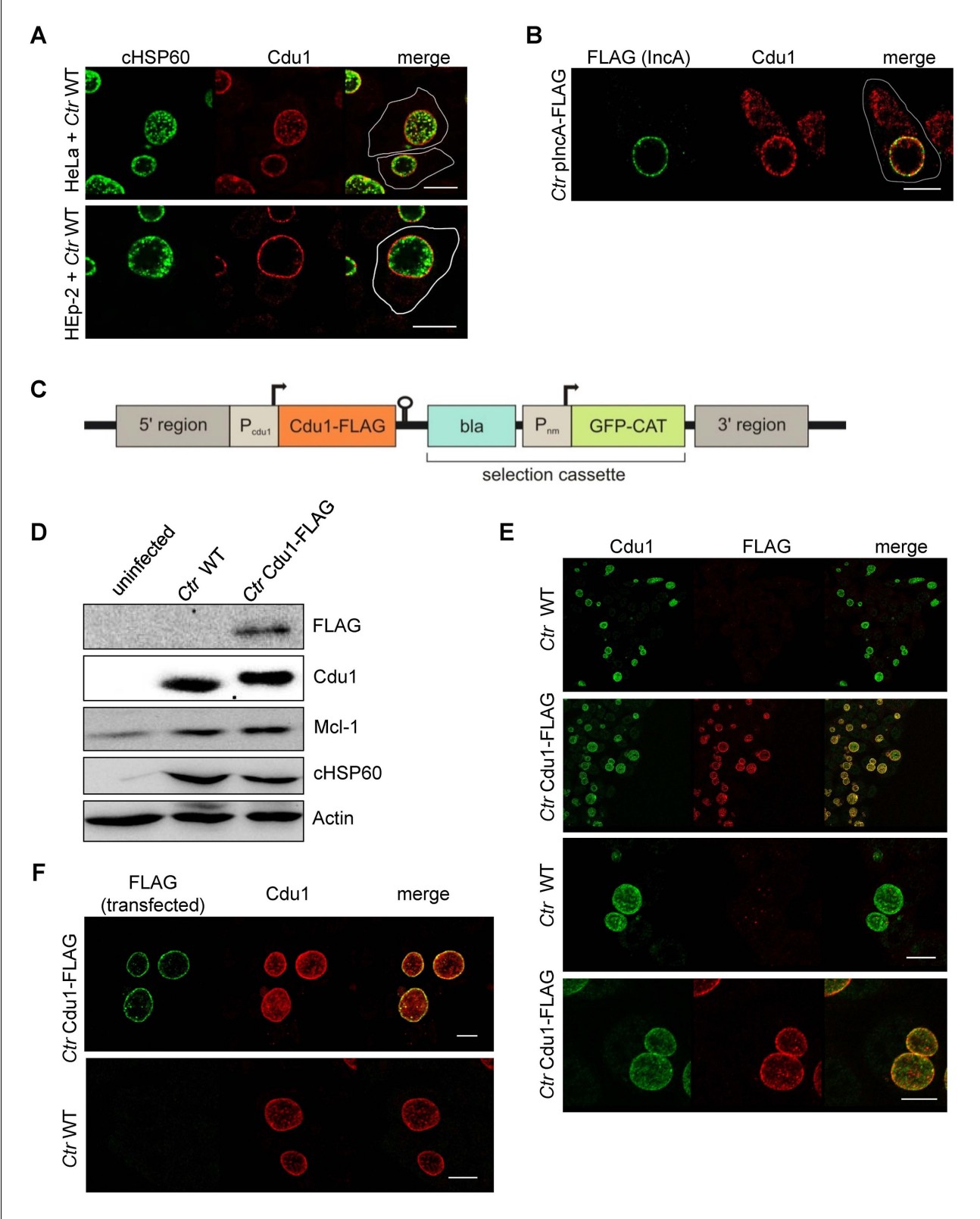

**Figure 3.** Subcellular localization of Cdu1 during infection. (**A**) HeLa (upper panel) and HEp-2 (lower panel) cells were infected with *C. trachomatis* (*Ctr*) at a MOI 1 for 24 hr and fixed with 4% PFA/Sucrose, followed by the indirect immunofluorescence staining against chlamydial HSP60 (green channel) and Cdu1 (red channel). Contours of cells are marked in white. Scale bar, 10 μm. (**B**) HeLa cells were infected for 24 hr with *Ctr* expressing an IncA-FLAG fusion protein. Cells were fixed with 4% PFA/Sucrose and indirect immunofluorescence staining against FLAG (IncA; green channel) and Cdu1

*Figure 3 continued on next page*

Figure 3 continued

(red channel) was performed. The contour of the cell is marked in white. The bar represents 10 µm. (**C**) Genomic region of the *cdu1* locus after recombination in the *Ctr* Cdu1-FLAG strain. (**D**) HeLa cells were infected with *Ctr* wild-type (WT) or *Ctr* Cdu1-FLAG for 24 hr at a MOI of 1. Whole cell lysates were prepared and immunoblot analysis of Cdu1 and indicated proteins was performed. (**E**) Immunofluorescence staining of HeLa cells infected with *Ctr* WT or *Ctr* Cdu1-FLAG. Staining was performed with an anti-Cdu1 (green channel) and an anti-FLAG (red channel) antibody. Scale bar, 10 µm. (**F**) HEK-293T cells grown on cover slips were infected with *Ctr* WT or *Ctr* Cdu1-FLAG. 20 hpi, cells were transfected with an anti-FLAG monoclonal antibody using the Chariot protein delivery reagent and fixed 4 hr post transfection. Transfected FLAG antibody was detected by indirect immunofluorescence with a Cy5-coupled anti-mouse secondary antibody (green channel) and Cdu1 was detected using the anti-Cdu1 antibody (red channel). Scale bar, 10 µm.

The following source data and figure supplements are available for figure 3:

**Figure supplement 1.** Generation and validation of the *C*.

**Figure supplement 2.** Characterization of *C. trachomatis* Cdu1-FLAG.

**Figure supplement 2—source data 1.** Raw data for quantitative analysis of *C. trachomatis* Cdu1-FLAG infectivity, Mcl-1 stabilization and apoptosis inhibition shown in *Figure 3—figure supplement 2*.

**Figure supplement 3.** Cdu1 expression in *C*.

**Figure supplement 4.** Cdu1 secretion in *C. trachomatis* Cdu1-FLAG.

transfected with the empty vector or inactive Cdu1(C345A)-FLAG compared to cells co-expressing Cdu1-FLAG (*Figure 5C*, left panel). The final proof that Cdu1 directly deubiquitinates Mcl-1 was provided in an in vitro deubiquitination assay. Purified recombinant Mcl-1 was poly-ubiquitinated in vitro and was subsequently used as substrate for purified recombinant Cdu1, its mutant Cdu1(C345A) or the unrelated deubiquitinase UCH-L3. Cdu1, but not the control enzyme UCH-L3 strongly deubiquitinated Mcl-1 (*Figure 5D*). Deubiquitination of Mcl-1 was dependent on its enzymatic activity, since neither the Cdu1(C345A) mutant nor the wild-type enzyme blocked with the universal cysteine protease inhibitor N-ethylmaleimide (NEM) deubiquitinated Mcl-1 (*Figure 5D*). Furthermore, poly-ubiquitinated proteins were isolated by TUBE pull down and incubated with Cdu1 or the control DUB USP28. Cdu1 specifically deubiquitinated Mcl-1 but not the control protein p53 (*Figure 5—figure supplement 1A*). This in vitro deubiquitination assays confirms that Mcl-1 is a substrate of Cdu1. It has already been demonstrated in vitro that Cdu1 is a deubiquitinase and deneddylase (*Misaghi et al., 2006*). Mcl-1 is primarily K48-ubiquitinated to target it to proteasomal degradation (*Mojsa et al., 2014*; *Wang et al., 2014*). We were able to show that Cdu1 cleaves K48- and K63-linked ubiquitin chains as it was recently described (*Pruneda et al., 2016*) (*Figure 5E*) supporting Mcl-1 being a substrate of Cdu1.

## A chlamydial Cdu1 mutant (*Ctr cdu1*::Tn *bla*) fails to deubiquitinate Mcl-1

Generating a chlamydial *cdu1* mutant by transforming a plasmid that should integrate the described selection cassette (*Figure 3C*) into the *cdu1* gene causing its disruption failed. In another approach, transposon mutagenesis using a *Himar1* transposon system was established in *C. trachomatis*. One of the resulting mutant strains (*Ctr cdu1*::Tn *bla*), termed *Ctr* Tn-*cdu1*, had a transposon inserted in the active site of Cdu1, resulting in a truncated version with 297 amino acids and an apparent molecular weight of 33 kDa (*Figure 6A and B*). To confirm the transposon insertion site and discover any other potential genetic differences that may contribute the phenotypes observed between *C. trachomatis* parental and Tn-*cdu1* mutant clones, genomic sequencing and analysis was performed. Comparison of the parental clone to the reference LGV 434/Bu genome revealed two mutations. One mutation (base 946,499) was in the coding region for CTL0818 (helicase) that resulted in an amino acid substitution (N1115H). The other was in CTL0817 (*brnQ*) and also resulted in a substitution (R275H). Interestingly, the reads for these mutations indicated a mixed population with 37% (CTL0818) and 65% (CTL0817) matching the reference strain sequence. Comparison of the Tn-*cdu1*

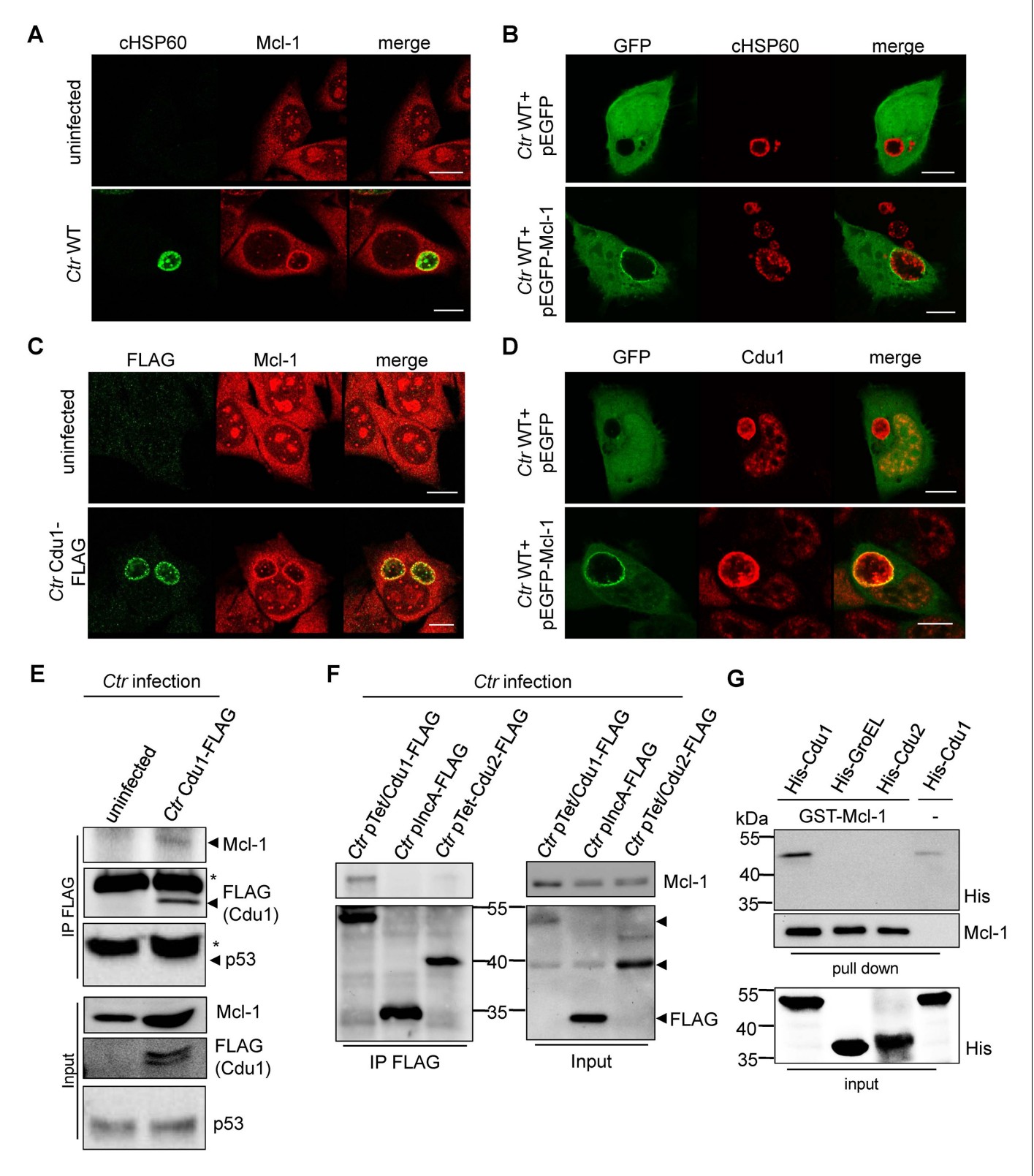

**Figure 4.** Cdu1 and Mcl-1 interact with each other. (**A**) Immunofluorescence staining of cellular Mcl-1 in uninfected and *C. trachomatis* (*Ctr*)-infected HeLa cells. Cells were stained with antibodies against chlamydial HSP60 (green channel) and Mcl-1 (red channel). Scale bars, 10 μm. (**B**) HeLa cells expressing EGFP or the EGFP-Mcl-1 fusion protein were infected with *Ctr* wild-type (WT) and fixed with 4% PFA/Sucrose 24 hpi. Chlamydial particles were stained with a cHSP60 antibody (red channel). Association of EGFP-Mcl-1 but not of EGFP with the chlamydial inclusion was visualized by confocal

*Figure 4 continued on next page*

*Figure 4 continued*

microscopy. Scale bar represents 10 µm. (C) HeLa cells were infected with *Ctr* WT or *Ctr* Cdu1-FLAG for 24 hr. Subcellular localization of Cdu1-FLAG (green channel) and Mcl-1 (red channel) was visualized. Bars represent 10 µm. (D) HeLa cells expressing EGFP or EGFP-Mcl-1 were infected with *Ctr* WT and fixed 24 hpi with 4% PFA/Sucrose. Cdu1 was marked by indirect immunofluorescence staining (red channel) and co-localization of EGFP or Mcl-1-EGFP (green channel) with Cdu1 was analyzed by confocal microscopy. Scale bar 10 µM. (E) Cdu1-FLAG was immunoprecipitated out of *Ctr* Cdu1-FLAG-infected HeLa and uninfected control cells using FLAG-tag directed antibodies. Co-precipitation of Mcl-1 and Cdu1-FLAG was shown by immunoblot. Probing with p53 antibody serves as specificity control. The asterisk marks the signal from the heavy chain of the IgG antibody. (F) Immunoprecipitation of Cdu1-FLAG out of HeLa cells infected with *Ctr* pTet/Cdu1-FLAG, *Ctr* pIncA-FLAG or *Ctr* pTet/Cdu2-FLAG. Co-precipitation of Mcl-1 and Cdu1-FLAG was visualized by immunoblot. (G) In vitro binding assay of recombinant purified GST-Mcl-1 and His-Cdu1. GST-Mcl-1 was bound to glutathione-sepharose beads and incubated with equal amounts of His-Cdu1 or control proteins His-Cdu2 and truncated His-GroEL. After washing, the interaction of the proteins was analyzed by immunoblot with antibodies detecting the His-tag or Mcl-1.

The following figure supplement is available for figure 4:

**Figure supplement 1.** Cellular Mcl-1 interacts with Cdu1.

genome to the parental clone verified that the transposon is located as indicated in *Figure 6A* and introduces a Y298* non-sense mutation in *cdu1*. Both sites in CTL0818 and CTL0817 that have mixed sequence in the parental strain, uniformly matched the reference strain; however, two new mutations were incurred in the Tn-*cdu1* strain. One mutation (base 929,586) occurred in the intergenic region between two diverging genes (CTL0805; *hisS* and CTL0806; *uhpC*) and a missense mutation (base 11,738) was found in the gene encoding for RecC (CTL0008) and results in an amino acid substitution (S638Y). Taken together, this analysis reveals that the genomes of the parent and Tn-*cdu1* strain are almost identical with the exception of the transposon insertions. While there are two distinct mutations in the Tn-*cdu1* clone relative to the parent clone, the only modification to a coding region was in RecC (S638Y). These data support that the phenotypes observed of the Tn-*cdu1* insertional mutant clone are almost definitively due to the functional disruption of the Cdu1 deubiquitinase.

Immunofluorescence analysis of HeLa cells infected with the *Ctr* Tn-*cdu1* mutant strain revealed reduced protein levels as well as impaired secretion of the truncated Cdu1 protein to the surface of the inclusion (*Figure 6C* and *Figure 6—figure supplement 1A and B*). However, a reduced Cdu1 signal can be explained by the usage of a polyclonal serum for Cdu1 detection and the deletion of potential immunogenic epitopes. Interestingly, Mcl-1 levels were reduced in cells infected with *Ctr* Tn-*cdu1* compared to cells infected with wild-type bacteria (*Figure 6B,D* and *Figure 6—figure supplement 1C*). However, the Mcl-1 level did not decrease to the level detected in uninfected control cells highlighting that *Chlamydia* established multiple mechanisms to induce Mcl-1 accumulation in the cell. Moreover, Mcl-1 was less accumulated around the inclusion of the *Ctr* Tn-*cdu1* mutant strain compared to wild-type *Chlamydia*-infected cells (*Figure 6E–F* and *Figure 6—figure supplement 2A*). Mcl-1 is still recruited to the inclusion of Tn-*cdu1* *Chlamydia*, however, the amount of inclusion-associated Mcl-1 is decreased in *Ctr* Tn-*cdu1*-infected cells compared to the WT (*Figure 6G*). Furthermore, chlamydial overexpression of Cdu1 resulted in a slight increase in Mcl-1 accumulation around the inclusion (*Figure 6—figure supplement 2B–C*).

The analysis of the ubiquitination pattern of Mcl-1 isolated from cells infected with wild-type or Tn-*cdu1* mutant *Chlamydia* clearly showed that Mcl-1 isolated from cells infected with the mutant bacteria is more ubiquitinated, in particular K48-linked ubiquitinated, compared to the *Ctr* wild-type-infected sample (*Figure 7A*, left panel and *Figure 7—figure supplement 1A and B*). However, Mcl-1 still shows reduced ubiquitination in comparison to the uninfected control, which could be a result of the sequestration of Mcl-1 to the inclusion (see *Figure 6E and F*). It is unclear if Mcl-1 is actively recruited to the inclusion or if it is accumulating by interaction with Cdu1. However, it would be expected that Mcl-1 present at the inclusion surface is more ubiquitinated in the mutant than in the wild-type. Indeed, immunofluorescence staining for ubiquitin or K48-linked ubiquitin and Mcl-1 in *Ctr* Tn-*cdu1*-infected cells shows that ubiquitin is accumulating around the inclusion in part co-localizing with Mcl-1 (*Figure 7B* and *Figure 7—figure supplement 1C–E*). Using structured illumination microscopy (SIM), we show that the inclusion-associated Mcl-1 is strongly ubiquitinated in *Ctr* Tn-*cdu1*-infected cells in comparison to cells infected with wild-type *Chlamydia*. The increase in the ubiquitination of Mcl-1 is exhibited by the rise in the degree of co-localization as shown by Pearson's

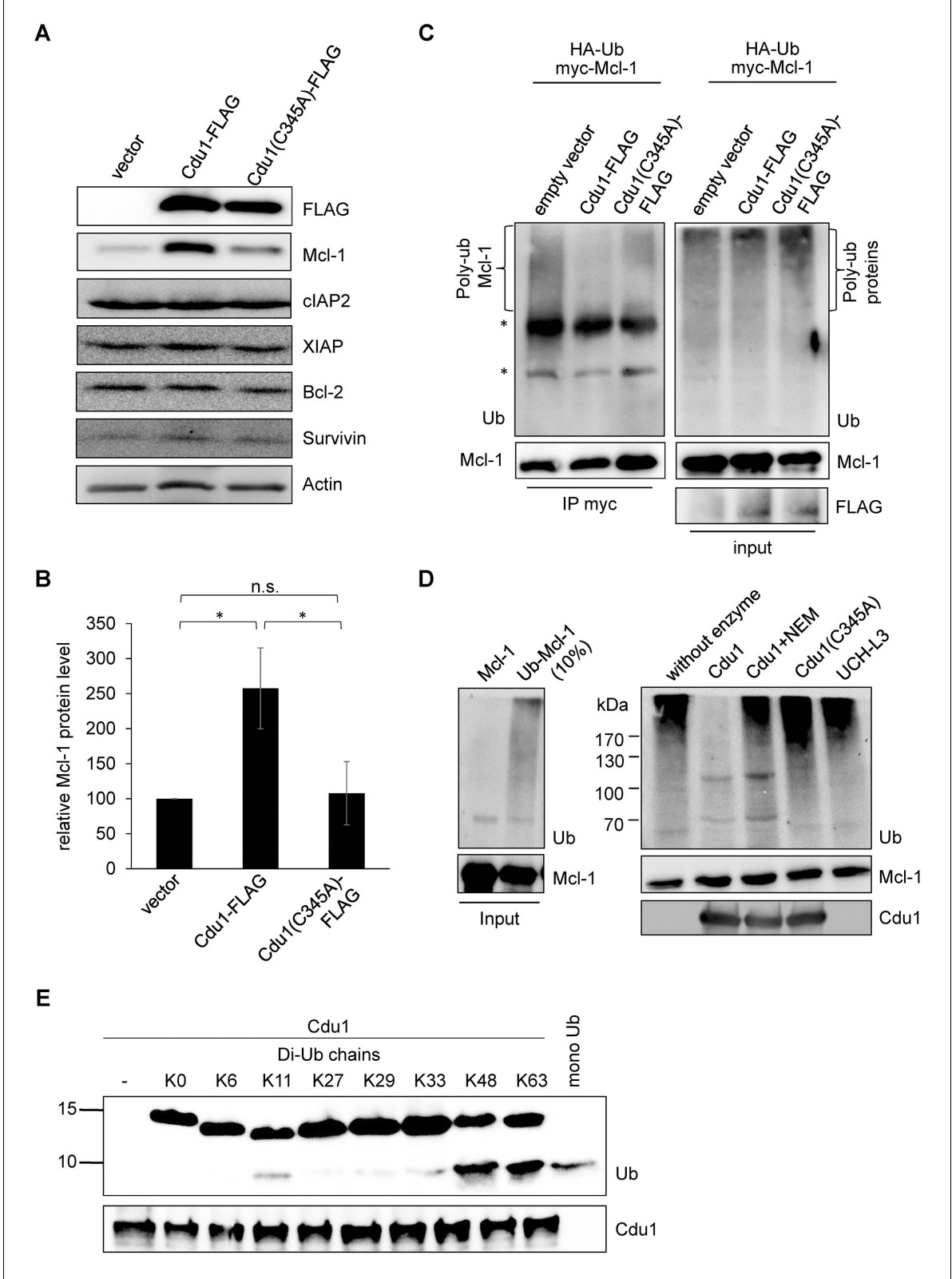

**Figure 5.** Cdu1 deubiquitinates Mcl-1 in vivo and in vitro. (**A**) HEK-293T cells were transfected with Cdu1-FLAG, Cdu1(C345A)-FLAG or pcDNA3 empty vector for 24 hr. Cells were lysed in SDS-sample buffer and protein expression as well as the effect on Mcl-1 or other anti-apoptotic proteins was analyzed by immunoblot. (**B**) Relative amount of Mcl-1 in HEK-293T cells expressing Cdu1, Cdu1(C345A) mutant or empty vector 24 hr after transfection calculated from experiment (**A**). Normalization was performed against an actin loading control. The graph shows mean values from three individual

*Figure 5 continued on next page*

*Figure 5 continued*

experiments ± SD. The significance was calculated with the student's T-test *p<0.05. (**C**) HEK-293T cells were transfected with HA-ubiquitin, myc-Mcl-1 and either empty vector control, Cdu1-FLAG or Cdu1(C345A)-FLAG for 24 hr. Transfected cells were treated with 10 µM MG132 over night to block proteasome activity and to enrich ubiquitinated proteins. Cells were lysed under denaturing conditions and Mcl-1 was immunoprecipitated by its myc-tag under stringent conditions. Ubiquitination status of all cellular proteins (input) and precipitated Mcl-1 was visualized by immunoblot using an anti-ubiquitin antibody. (**D**) Purified recombinant Mcl-1 was ubiquitinated in vitro (left panel). Poly-ubiquitinated Mcl-1 served as a substrate in an in vitro DUB-assay for the purified recombinant enzymes Cdu1, inactive Cdu1 or an unrelated control deubiquitinase UCH-L3. The ubiquitination status of Mcl-1 was analyzed by immunoblot using an anti-ubiquitin antibody. (**E**) Ubiquitin chain specificity of Cdu1. 0.5 µg di-ubiquitin chains were incubated with 20 nM Cdu1 for 2 hr at 37°C, and chain cleavage was analyzed by immunoblot using an anti-ubiquitin antibody. See also *Figure 5—source data 1*.

The following source data and figure supplement are available for figure 5:

**Source data 1.** Raw data for quantitative analysis of Mcl-1 stabilization in Cdu1-expressing HEK 293T cells shown in *Figure 5*.
**Figure supplement 1.** Cdu1 deubiquitinates Mcl-1 in vitro.

co-localization analysis (*Figure 7C* and *Figure 7—figure supplement 2A and B*). A 3D reconstruction of a *Ctr* Tn-*cdu1*-infected cell stained for Mcl-1 and ubiquitin reveals the special organization on the inclusion surface (*Figure 7—figure supplement 3* and *Videos 1–3*).

We challenged the transposon mutant *Chlamydia* with TNFα/CHX in an apoptosis assay, however, the Cdu1 mutant did not show any sensitization towards apoptosis induction compared to the wild-type *Chlamydia* (*Figure 7D*). In cells infected with *Ctr* Tn-*cdu1* the MEK/ERK survival signaling pathway was still activated and Mcl-1 levels were increased which may compensate for the lack of Mcl-1 stabilization by Cdu1 (*Figure 6B and D*). On the contrary, the *Ctr* Tn-*cdu1* mutant strain showed a sensitization toward immune stimulation with interferon (IFN) γ, which became evident by a significant reduction of replication in primary cells of human fimbriae obtained from biopsies (referred as Fimb cells) reflecting the natural infection site for *C. trachomatis* after IFNγ treatment (*Figure 7E*).

To evaluate the potential importance of Cdu1 during mammalian infection, mice were challenged with *Ctr* Tn-*cdu1* using a trans-cervical infection method (*Gondek et al., 2012*). In addition to wild-type *C. trachomatis* (WT), mice were also infected with transposon mutant strain termed IGR::Tn *bla* (Tn-IGR). This mutant has a transposon inserted in the intergenic region between two converging genes (CT383/384) and is serving as a genetic control for potential fitness effects contributed by any transposon content including the *bla* resistance gene. The control *C. trachomatis* strain Tn-IGR was also sequenced and the location of the transposon between CTL0639/40 was confirmed. Sequencing also revealed that the missense mutation in CTL0818 (N1115H) of the parental strain was homogenously present, although the sequence for CTL0817 matched the reference sequence.

Nine mice were each challenged with $5 \times 10^5$ IFU and on day 5, uterine horns were extracted and bacterial burden determined by measuring the ratio of genome copies of *Chlamydia* relative to host (*Figure 7F*). Statistical analysis of the bacterial burdens of WT to Tn-IGR infected mice revealed no significant difference with mean genome ratios of $5.4 \times 10^{-3}$ and $4.4 \times 10^{-3}$, respectively. In contrast, statistical analysis revealed a highly significant difference (p<0.01) in bacterial burdens with over a log fold decrease in mean bacterial burden ($2.0 \times 10^{-4}$) of Tn-*cdu1*. These observations support the hypothesis that Cdu1 is important during mammalian infection.

## Discussion

The chlamydial inclusion is the interface to the eukaryotic host and serves as a signaling platform that orchestrates the accommodation of this obligate intracellular bacterium inside the host cell. Numerous secreted chlamydial proteins decorate the inclusion surface and function as scaffolds that recruit host trafficking and signaling molecules to the inclusion (*Elwell et al., 2016*). We demonstrate here the localization of the chlamydial deubiquitinating enzyme Cdu1 to the inclusion surface and its essential role for chlamydial development in IFNγ-challenged cultured cells and in a mouse infection model. This is the first report on a functionally and structurally eukaryotic-like enzyme encoded and secreted by *Chlamydia* that is required for the adaptation and survival of these bacteria. We further

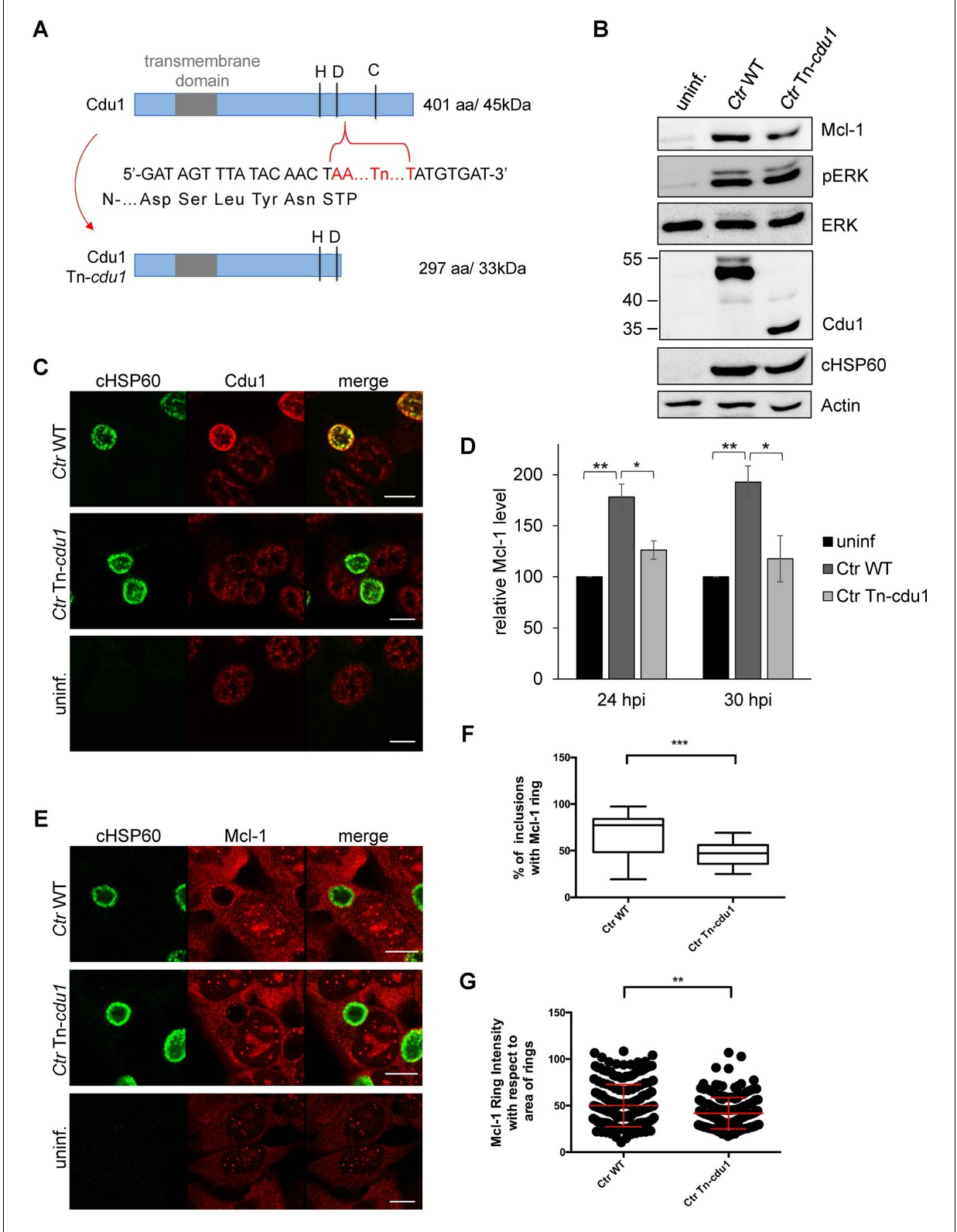

**Figure 6.** The Cdu1 truncated mutant affects Mcl-1 stabilization. (**A**) Scheme of the Cdu1 protein in wild-type (WT) *Chlamydia* and the transposon mutant (Tn-*cdu1*) lacking the catalytically active cysteine 345. (**B**) Immunoblot analysis of HeLa cells infected for 24 hr with WT or Tn-*cdu1* mutant *C. trachomatis* (*Ctr*) with the indicated antibodies. Decoration with cHSP60 and Actin serves as infection and loading control. (**C**) HeLa cells were infected with *Ctr* WT and Tn-*cdu1* mutant and fixed 24 hpi with 4% PFA/Sucrose. Indirect immunofluorescence staining for cHSP60 (green channel) and Cdu1

*Figure 6 continued on next page*

*Figure 6 continued*

(red channel) was performed. The Cdu1 antibody shows unspecific staining of nuclear structures. Scale bar 10 μm. (**D**) Relative amount of Mcl-1 in HeLa cells infected with *Ctr* WT and Tn-*cdu1* mutant for 24 or 30 hr. Normalization was performed against an Actin loading control. The graph shows mean values from three individual experiments ± SD. The significance was calculated with the student's T-test *p<0.05, **p<0.01. (**E**) Indirect immunofluorescence staining for Mcl-1 (red channel) of HeLa cells infected with *Ctr* WT or the Tn-*cdu1* mutant for 24 hr. Chlamydial particles were marked with cHSP60 antibody (green channel). Scale bar represents 10 μm. (**F+G**) Analysis of inclusion-associated Mcl-1 in HeLa cells infected with *Ctr* WT or Tn-*cdu*1. Infected HeLa cells were fixed 24 hpi and immunostaining against Mcl-1 and cHSP60 was performed. Relative number of Mcl-1-decorated inclusions of WT or Tn-*cdu1 Chlamydia* were calculated. The graph shows mean values from three individual experiments ± SD. The significance was calculated with the student's T-test ***p<0.001. The source data for number of Mcl-1 decorated *Ctr* WT inclusions matches with source data from *Figure 4—figure supplement 1C* (**F**). Quantification of inclusion-associated Mcl-1 in *Ctr* WT or Tn-*cdu1*-infected cells. The graph shows mean values from three individual experiments ± SD. The significance was calculated with the Kolmogorov-Smirnov test; **p<0.01 (**G**). See also *Figure 6—source data 1*.

The following source data and figure supplements are available for figure 6:

**Source data 1.** Raw data for quantitative analysis of Mcl-1 stabilization, Mcl-1-ringed inclusion count and calculation of Mcl-1 ring intensity of HeLa cells infected with *C. trachomatis* wild type and Tn-*cdu1* shown in *Figure 6*.
**Figure supplement 1.** Characterization of *C. trachomatis* Tn-cdu1 mutant.
**Figure supplement 2.** Mcl-1 association with the chlamydial inclusion.
**Figure supplement 2—source data 1.** Raw data for quantitative analysis of Mcl-1 ring intensity around *C. trachomatis* pTet/Cdu1 inclusions uninduced and induced for Cdu1 overexpression shown in *Figure 6—figure supplement 2*.

show that Cdu1 specifically deubiquitinates the major apoptosis regulator Mcl-1 at the surface of the inclusion.

Due to the high turn-over rate of Mcl-1, actively growing *Chlamydia* have to interfere with a very well-balanced cellular system consisting of several Mcl-1-specific ubiquitin ligases and the deubiquitinating enzyme USP9X (*Schwickart et al., 2010*; *Zhong et al., 2005*; *Thomas et al., 2010*) that constantly tune the levels of Mcl-1. Early in infection, expression of Mcl-1 is increased by the activation of the RAF/MEK/ERK and the MAPK/AKT signaling pathway (*Rajalingam et al., 2008*). Maintenance of increased Mcl-1 levels in the mid and late phases of infection, however, required in addition stabilization of the protein. We show here that Mcl-1 stabilization is a result of reduced ubiquitination and proteasomal degradation induced by the chlamydial Cdu1 that is independent of the known major host UPS factors regulating Mcl-1 stability.

Mcl-1 may be just one of several Cdu1 targets that are deubiquitinated in infected cells. Furthermore, with the exception of Cdu2, Cdu1 is the only known bacterial DUB that possesses both deubiquitinating and deneddylating activities (*Misaghi et al., 2006*) indicating that also Cdus' deneddylating activity may be relevant for the intracellular adaptation of the pathogen. The structural similarity between Cdu1, Ulp1 and SENP8 with respect to the position of the catalytic triad suggests that Cdu1 should bind ubiquitin and Nedd8 in a similar way as observed for the Ulp1-SMT3 and SENP8-Nedd8 complexes. A major structural difference, however, is the additional α-helix D in Cdu1, which is not present in the other enzymes, and its proximity to the Ub/Nedd8 C-terminus. We speculate that this helix could act like a 'lid' that plays a role in substrate binding and/or recognition and assumes a different orientation when the substrate is bound.

Defining precisely the intracellular localization of Cdu1 in infected cells surrounding the inclusion was of major importance to understand the mechanism of Mcl-1 stabilization. Although it was not possible to detect secreted Cdu1 in the cytosol, it cannot be excluded that a minor fraction of Cdu1 that escaped detection in the here described approaches is also present in the cytosol of infected host cells, as has previously been described (*Le Negrate et al., 2008*). Nevertheless, the localization of a major fraction of Cdu1 in the inclusion membrane of cells infected with wild-type and to a lesser extend with *Ctr* Tn-*cdu1* coincides with the high enrichment of Mcl-1 in the immediate vicinity of the Cdu1-containing compartment. Enrichment of high amounts of Mcl-1 around the inclusion may result from protein stabilization by Cdu1 in this defined area. However, Mcl-1 may also be actively recruited to the inclusion since the interaction of Cdu1 with endogenous Mcl-1 was shown in living

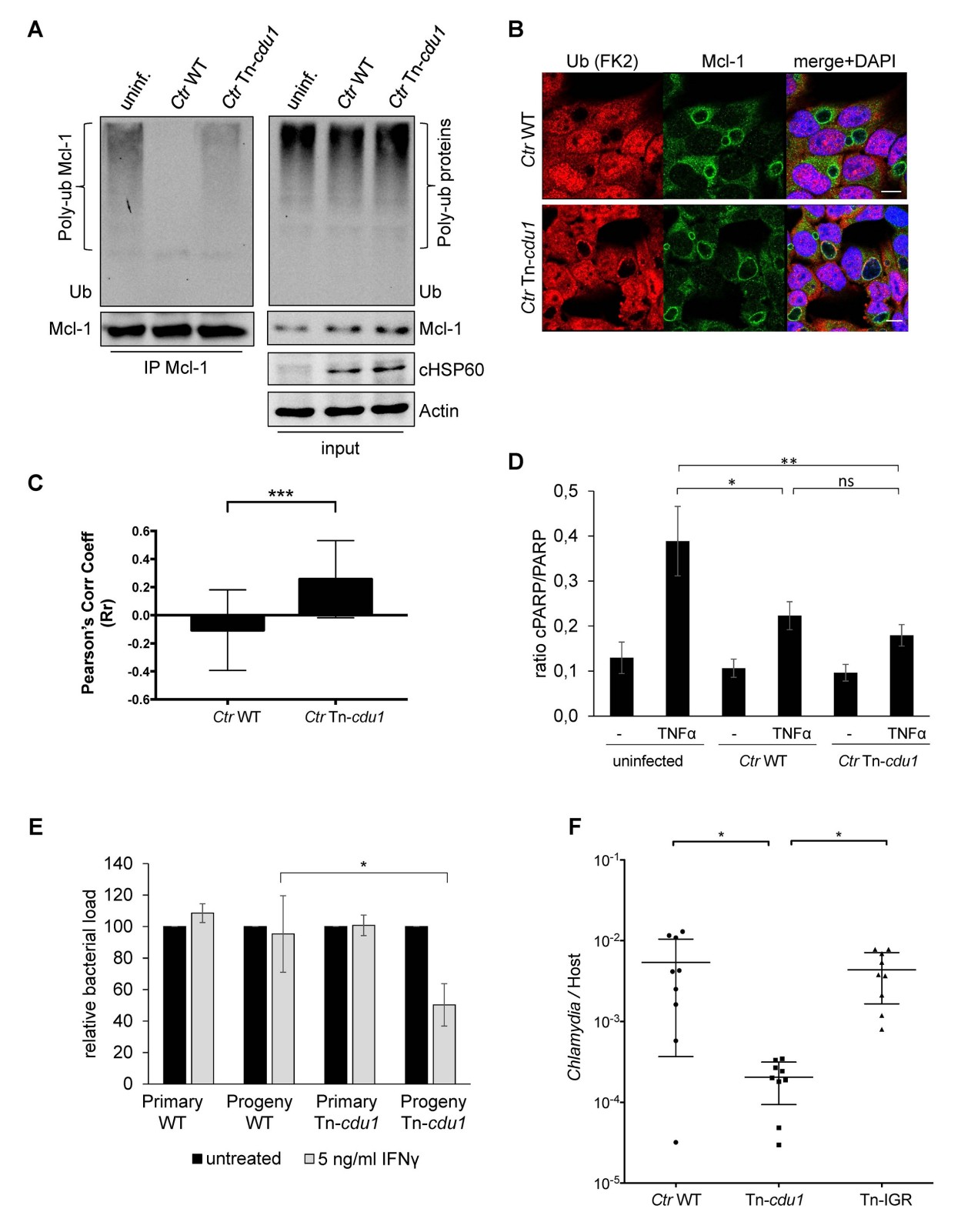

**Figure 7.** Disruption of *cdu1* affects Mcl-1 ubiquitination and survival of *Chlamydia*. (**A**) HeLa cells were infected with *C. trachomatis* (*Ctr*) wild-type (WT) or Tn-*cdu1* and the proteasome inhibitor MG-132 was added 8 hpi. Cells were lysed under denaturing conditions 24 hpi and Mcl-1 was precipitated. Precipitated Mcl-1 (left panel) as well as the whole proteome (right panel) were analyzed for their ubiquitination pattern by immunoblot using an anti-ubiquitin antibody. (**B**) HeLa cells were infected with *Ctr* WT or the Tn-*cdu1* mutant for 24 hr and fixed with 4% PFA/Sucrose. Indirect

*Figure 7 continued on next page*

Figure 7 continued

immunofluorescence staining against ubiquitin (FK2, red channel) and Mcl-1 (green channel) was performed. DAPI staining (blue channel) marks the host cell nucleus and chlamydial DNA. (C) Quantification of ubiquitin co-localization with Mcl-1 by comparison of Pearson's co-localization coefficient using the COLOC2 plugin from FIJI. 10 ROI were randomly selected from five individual experiments. The significance was calculated with the Student's T-test ***p<0.001. (D) HeLa cells were infected with *Ctr* WT or Tn-*cdu1*. Apoptosis was induced 20 hpi with 50 ng/ml TNFα and 5 µg/ml CHX. Apoptosis induction was analyzed by PARP cleavage in an immunoblot. Depicted are mean values from the ratio of cleaved PARP/PARP calculated from four individual experiments ± SD. The significance was calculated with the student's T-test *p<0.05, **p<0.01. (E) Infectivity assay of *Ctr* WT and Tn-*cdu1* in primary Fimb cells. Infected cells were challenged with 5 ng/ml IFNγ upon 8 hpi. Depicted are mean values of the bacterial load calculated from three individual experiments ± SD. The significance was calculated with the student's T-test *p<0.05. (F) Female C57BL/6 mice (n = 9) were infected transcervically with either *Ctr* WT, Tn-*cdu1* or Tn control Tn-IGR. Five days post-infection, genital tracts were harvested, homogenized and DNA isolated. DNA from uterine horns was subjected to ddPCR to determine detectable *Chlamydia* copies/µL. Samples are normalized to host DNA (copies/µL). WT and Tn-IGR-infected animals showed no significant difference while both WT vs Tn-*cdu1* and Tn-IGR vs Tn-*cdu1* showed a significant difference between detectable copies/µL (*p<0.01, Kruskall-Wallis test with Dunn's multiple comparison post-test). See also *Figure 7—source data 1*.

The following source data and figure supplements are available for figure 7:

**Source data 1.** Raw data for quantitative analysis of apoptosis resistance analyzed by PARP cleavage and analysis of infectivity after IFNγ-treatment shown in *Figure 7*.

**Figure supplement 1.** Ubiquitin is accumulated around the inclusion of Tn-*cdu1* mutant *Chlamydia*.

**Figure supplement 2.** SIM analysis of Mcl-1 ubiquitin co-localization at the inclusion surface.

**Figure supplement 3.** 3D reconstruction of a Mcl-1/Ubiquitin co-staining.

cells. Additionally, the purified recombinant proteins bound to each other in in vitro binding assays suggesting that they interact directly and no additional host factors are required for their interaction outside the inclusion. Since previous work on the cellular Mcl-1 deubiquitinase USP9X demonstrated that deubiquitinase and target do not have to reside in the same cellular compartment (*Schwickart et al., 2010*) it can be speculated, that inclusion-associated Cdu1 constantly increases the pool of cytosolic Mcl-1 and thereby also the amount of effective anti-apoptotic Mcl-1 at the mitochondria. Our co-localization analysis using SIM clearly demonstrated reduced ubiquitinated Mcl-1 at the inclusion surface of wild-type *Chlamydia* compared to inclusions of the Tn-*cdu1* mutant strain (*Figure 7C*).

Here, the targeted recombination into the chlamydial genome based on a newly established protocol for the transformation of *C. trachomatis* with a suicide plasmid (*Wang et al., 2011*) is described. Although the integration of a FLAG-tagged version of Cdu1 into the *C. trachomatis* genome replacing the wild-type gene was effective, generating a knock-out by deleting or interrupting the *cdu1* gene close to the N-terminus was not successful. The isolation of Tn-*cdu1* demonstrated that at least the catalytic activity of Cdu1 is not essential for chlamydial growth in cultured cells.

Mouse infection investigations (*Figure 7F*) support that Cdu1 is important to mammalian infection. While the systems for genetic manipulation have been in development recently, this mouse challenge study presented herein is among the only animal challenge study that has been performed with genetically manipulated *Chlamydia* strains. Moreover, the genetic disruption resulted in a substantial and significant

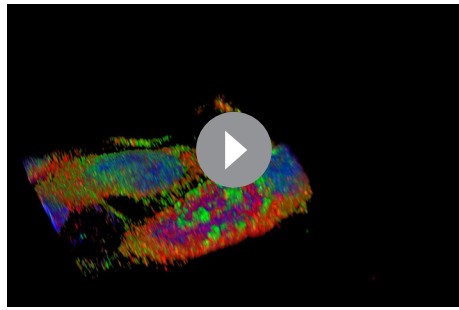

**Video 1.** 3D reconstruction of a Mcl-1/Ubiquitin co-staining. Diagonal rotation around Y of the 3D projection of a HeLa cell infected with *C. trachomatis* Tn-*cdu1*. Staining was performed against Mcl-1 (red channel), ubiquitin (green channel) and DNA is stained with DAPI. The ubiquitin signal (green) can be seen to co-localize with Mcl-1 (red) on the surface of the inclusion (purple arrows). Snapshots are depicted in *Figure 7—figure supplement 2*.

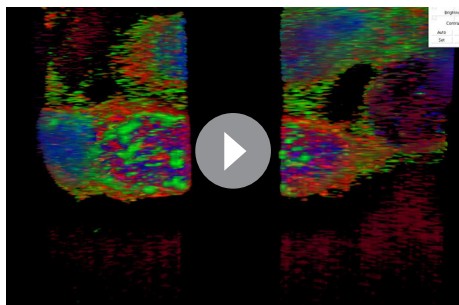

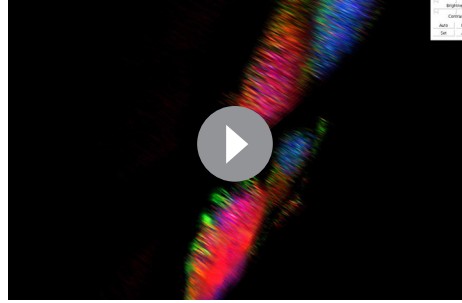

**Video 2.** 3D reconstruction of a Mcl-1/Ubiquitin co-staining. Diagonal rotation of the 3D projection around Y of Tn-*cdu1 C. trachomtis*-infected HeLa cells stained with Mcl-1 (red channel), ubiquitin (green channel) and DNA is stained with DAPI (blue channel). Blank stacks were inserted in-between to generate a lateral cross-section of the chlamydial inclusion.

**Video 3.** 3D reconstruction of a Mcl-1/Ubiquitin co-staining. Straight rotation around Y of the 3D projection of HeLa cells infected with *C. trachomatis* Tn-*cdu1*. Immunofluorescence staining was performed against Mcl-1 (red channel), ubiquitin (green channel) and DAPI staining marks the DNA (blue channel). Blank stacks were inserted in-between to generate a lateral cross-section of the chlamydial inclusion.

effect on infection with over a log-fold decrease in bacterial burden. While it is unknown what specific aspect of infection is affected by this genetic disruption, a polar effect of the transposon insertion on genes downstream of the Cdu1 coding gene is unlikely, since *cdu2* (Cdu2) is directly downstream of *cdu1* and has been shown to have its own transcriptional start site (*Albrecht et al., 2010*). Further, Cdu2 does not appear to play a role with Mcl-1 function since it failed to interact with Mcl-1 (*Figure 4F and G*) and Mcl-1 still localizes to the inclusion in the *cdu1* allelic replacement strain (*Ctr* Cdu1-FLAG). Therefore, the loss of Mcl-1 interaction with the chlamydial inclusion and functional effect due to higher ubiquination levels is likely maintained during mammalian infection by the Tn-*cdu1* mutant strain. It is important to appreciate that these observations support that *Chlamydia*-directed deubiquitination activities appear to have a substantial effect on mammalian infection.

# Materials and methods

## Cell culture and transfection

HeLa229 (ATCC Cat# CCL-2.1, RRID:CVCL_1276; tested negative for mycoplasma contamination by PCR), HEp-2 (ATCC Cat# CCL-23, RRID:CVCL_1906; tested negative for mycoplasma contamination by PCR), McCoy (ATCC Cat# CRL-1696, RRID:CVCL_3742; tested negative for mycoplasma contamination by PCR) and primary Fimb (primary cells of human fimbriae obtained from biopsies of patients undergoing medically indicated surgery. According to the hospitalization contract patients agreed to scientific use of biopsies after anonymization of samples (tested negative for mycoplasma contamination by PCR); cells were cultured in RPMI-1640 medium (Gibco, Waltham, MA) and 293T cells (ATCC Cat# CRL-3216, RRID:CVCL_0063; tested negative for mycoplasma contamination by PCR) in Dulbecco's Modified Eagle Medium (DMEM) (Gibco). The medium was supplemented with 10% fetal calf serum (FCS) (PAA, Germany) and cells were incubated at 37°C in 5% $CO_2$. Cells were transfected with plasmid DNA at a confluency of 60% with Polyethylenimine (PEI) and OptiMEM transfection medium (Gibco) in 5% FCS medium. After 5 hr, transfection medium was replaced by fresh RPMI supplemented with 5% FCS medium. Plasmids used in this study are described in *Supplementary file 2*. The pCDNA3/HA-Ubiquitin construct was a kind gift of D. Bohmann (*Treier et al., 1994*), the pCDNA3.1/hMcl-1 was provided by the Roger Davis lab (*Morel et al., 2009*) and the pGEX4t3/ΔNHECTH9 construct was a gift of Martin Eilers (*Adhikary et al., 2005*). Protein transfection of *C. trachomatis*-infected 293T cells was performed at a confluency of 70% using the Active Motif Chariot kit following the manual.

## Growth of *Chlamydia*

Infection and stock preparation of *C. trachomatis* was performed as previously described (*Al-Younes et al., 1999*; *Böhme et al., 2010*). Cells were infected at a confluency of 70% with a MOI 1 in RPMI containing 5% FCS, infected cells were cultured at 35°C and 5% $CO_2$. After 2.5 hpi, the infection medium was replaced by fresh RPMI containing 5% FCS. For infection time course experiments, infection medium was replaced by fresh RPMI containing 5% FCS 1 hpi. To induce chlamydial protein overexpression in pTet-*Chlamydia* 10 ng/ml AHT was added to the infected cells. Apoptosis was induced 20 hpi with 50 ng/ml TNFα (Cell Signaling, Danvers, MA) in the presence of 5 µg/ml cycloheximide (CHX, Sigma Aldrich, Germany). The MEK/ERK inhibitor UO126 was added to the cells 12 hpi in the indicated concentration. *C. trachomatis* strains used in this study are listed in *Supplementary file 3*.

## Transformation of *C. trachomatis*

The transformation of *C. trachomatis* L2/434/Bu was performed as described by *Wang et al. (2011)*. In brief, plasmid DNA (10 µg) and 16 × 10⁶ IFU were mixed in $CaCl_2$ buffer and incubated for 20 min at RT. In the meantime, 8 × 10⁶ McCoy cells were washed with PBS and resuspended in $CaCl_2$ buffer. Cells and transfection mixture were combined and incubated another 20 min before cells were added in a T75 flask with fresh RPMI supplemented with 10% FCS. After 48 hr, the cells were lysed by sterile glass beads and the supernatant was used for infection of fresh McCoy cells. *Chlamydia*-infected cells were cultured in the presence of 2 U/ml Penicillin G and 2 µg/ml CHX and passaged every 48 hr. Upon active inclusions were visible, the concentration of Penicillin was increased up to 20 units. The recombinant *Chlamydia* strains created by this method are listed in *Supplementary file 3* . The pGFP::SW2 plasmid was a kind gift of I. Clarke (*Wang et al., 2011*).

## Infectivity assay

Cells were infected with the indicated *C. trachomatis* strain at a MOI 1. IFNγ (5 ng/ml) (Cell Signaling) was added to the cells 8 hpi. One set of the primary infection was lysed 24 hpi in 2x SDS-sample buffer. The second set of primary infection was lysed by glass beads 48 hpi. With this lysate fresh cells were infected for 24 hr before the progeny infection was lysed in 2x SDS-sample buffer. The degree of infection was analyzed by immunoblot detecting chlamydial HSP60.

## SDS-PAGE and immunoblotting

Protein samples were mixed with 2x SDS-sample buffer [100 mM Tris/HCl (pH 6.8), 20% glycerin, 4% SDS, 1.5% 2-mercaptoethanol, 0.2% bromphenolblue] and boiled at 94°C for 5 min. Samples were separated by 6–12% SDS-PAGE or Tricin-SDS-PAGE and transferred to a PVDF membrane (Roche) in a semi-dry electroblotter (Thermo Fisher Scientific). For immunoblot analysis, membranes were blocked with Tris-buffered saline containing 0.05% Tween20 and 5% bovine serum albumin or 5% dry milk powder. The following primary antibodies were used: anti-ubiquitin (Santa Cruz Biotechnology Cat# sc-8017 RRID:AB_628423), anti-ubiquitin FK2 (Enzo Life Sciences Cat# BML-PW8810 RRID:AB_10541840), anti-ubiquitin Lys48-specific (Millipore Cat# 05–1307 RRID:AB_1587578), anti-Mcl-1 (Abcam Cat# 1239–1 RRID:AB_562163), anti-Mcl-1 (Santa Cruz Biotechnology Cat# sc-819 RRID:AB_2144105), anti-myc (Santa Cruz Biotechnology Cat# sc-40 RRID:AB_627268), anti-chlamydial HSP60 (Santa Cruz Biotechnology Cat# sc-57840 RRID:AB_783868), anti-chlamydial HSP60 (rabbit antiserum), anti-β-Actin (Sigma-Aldrich Cat# A5441 RRID:AB_476744), anti-Cdu1, anti-His (Santa Cruz Biotechnology Cat# sc-8036 RRID:AB_627727), anti-FLAG (Sigma-Aldrich Cat# F3165 RRID:AB_259529), anti-cIAP2, anti-XIAP, anti-Bcl-2, anti-Survivin, anti-Ureb1/Lasu1 (Bethyl Labs Cat# A300-486A also ENCAB261PLB RRID:AB_2615536), anti-USP9X (Santa Cruz Bioscience), anti-PARP1/2 (Santa Cruz Biotechnology Cat# sc-7150 RRID:AB_2160738), anti-Fbw7, anti-Cul1 (Epitomics), anti-ERK (Cell Signaling Technology Cat# 9108 RRID:AB_2141156), anti-pERK (Cell Signaling Technology Cat# 9106 also 9106L, 9106S RRID:AB_331768), anti-p53 (Santa Cruz Biotechnology Cat# sc-6243 RRID:AB_653753) and anti-Bim (Epitomics). Proteins were detected with HRP-coupled secondary antibodies (Santa Cruz Bioscience) using the ECL system (Pierce) and Intas Chem HR 16–3200 reader. Quantification was done by the ImageJ software (http://imagej.nih.gov/ij; RRID:SCR_003070).

## Indirect immunofluorescence

HeLa229 and HEp-2 cells were grown on cover slips and infected with the indicated *C. trachomatis* strain at a MOI 1. The *C. trachomatis* pIncA-FLAG strain was a kind gift of P. Subbarayal (*Subbarayal et al., 2015*). At indicated time points, the cells were washed with PBS, fixed with 4% PFA/Sucrose and permeabilized with 0.2% Triton-X-100/PBS for 30 min. Samples were blocked with 2% FCS/PBS for 1 hr. All primary antibodies were incubated for 1 hr at RT. Primary antibodies were used in the following dilutions in 2% FCS/PBS: anti-HSP60 (Santa Cruz Biotechnology Cat# sc-57840 RRID:AB_783868; 1:300), anti-Mcl-1 (Santa Cruz Biotechnology Cat# sc-819 RRID:AB_2144105; 1:300), anti-FLAG (Sigma-Aldrich Cat# F3165 RRID:AB_259529; 1:300), anti-Cdu1 (1:200), anti-Ub-FK2 (Enzo Life Sciences Cat# BML-PW8810 RRID:AB_10541840; 1:100), anti-ubiquitin Lys48-specific (Millipore Cat# 05–1307 RRID:AB_1587578; 1:500). Samples were washed three times and incubated with a Cy2-/Cy3-/Cy5-conjugated secondary antibody for 1 hr in the dark.

## Immunofluorescence microscopy and image processing

Samples were imaged on a Leica SPE (RRID:SCR_002140) or a Leica TCS SP5 confocal microscope using a 63x oil immersion UV objective with a numerical aperture of 1.4. For super resolution imaging of Mcl-1-ubiquitin co-localization, images were acquired using a Zeiss ELYRA S.1 SR-SIM structured illumination platform using a Plan-APOCHROMAT 63x oil immersion objective with a numerical aperture of 1.4. The images were reconstructed using the ZEN 2012 image-processing platform with a SIM module. All image-processing steps were performed using FIJI (*Schindelin et al., 2012*) (RRID:SCR_002285). Mcl-1 co-localization with the ubiquitin signal was determined using the COLOC2 plugin from FIJI (*Bolte and Cordelières, 2006*). The degree of co-localization was ascertained using Pearson's co-localization coefficient.

3D reconstruction was rendered from Z-stacks of SIM images using the 3D viewer plugin from FIJI (*Schmid et al., 2010*). Briefly, the stacks were re-sliced in a top-bottom or left-right manner as per requirement followed by 3D rendering. Blank sections were introduced between the slides to generate the lateral cross-sections to provide an internal view.

The count of rings were done using the cell counter plugin (FIJI) followed by automated counting of the cell nucleus (stained with DAPI) and the number of chlamydial inclusions (stained against cHSP60) using the object count plugin form (FIJI). The percentage of chlamydial inclusions decorated with rings was calculated by diving the number of rings by the total number of chlamydial inclusions observed in each picture. Intensity of the Mcl-1 rings decorating chlamydial inclusions in wild-type (WT), Tn-*cdu1*, pTet/Cdu1 *Chlamydia* (with and without AHT treatment) were calculated using FIJI after appropriate thresholding for background. Briefly, ~20 rings were selected in random from several regions of interest (in three different samples for wild-type (WT) vs.Tn-*cdu1* and 2 different samples for pTet/Cdu1 *Chlamydia* with vs. without AHT treatment) and were measured for area and intensity of the Mcl-1 ring. The absolute intensity was calculated by dividing the observed arbitrary intensity value by the area of the ring which was then plotted to produce the graphs.

## Immunoprecipitation

For immunoprecipitation of Mcl-1, cells were pre-incubated with MG132 (as indicated) before lysis. To avoid protein co-precipitation with interaction partners cell lysates were prepared in RIPA-lysis buffer [50 mM Tris-HCl pH 7.5, 150 mM NaCl, 1% Triton X-100, 1 % NP-40, 0.1% SDS, 10% Glycerol] containing Complete protease inhibitor cocktail (Roche), 30 mM N-ethylmaleimide (NEM) and 30 µM MG132 proteasome inhibitor. $7 \times 10^6$ cells were lysed for 30 min at 4°C and debris was removed by centrifugation at maximum speed for 5 min. For denaturing lysis, cells were collected in 1% SDS lysis buffer [50 mM Tris-HCl pH 7.5, 1% SDS] and boiled for 10 min at 94°C. Lysate was diluted with nine volumes of RIPA-lysis buffer supplemented with Complete protease inhibitor cocktail (Roche) and 30 µM MG132 to a final SDS concentration of 0.1% and cellular debris was removed by centrifugation. Samples were incubated with 1:100 anti-Mcl-1 (Abcam Cat# 1239–1 RRID:AB_562163) or 1:100 anti-myc-tag (Santa Cruz Biotechnology Cat# sc-40 RRID:AB_627268) for 1 hr at 4°C. Protein G magnetic beads (Dynabeads, Thermo Fisher Scientific, Waltham, MA) were added and incubated for 2 hr at 4°C. After extensive washing, antigen-antibody complexes were eluted by boiling at 95°C in 2x SDS-sample buffer.

Lysates for co-immunoprecipitation of Cdu1-FLAG or Mcl-1 were prepared in Co-IP lysis buffer [10 mM Tris-HCl pH 7.5, 200 mM NaCl, 0.5 mM EDTA, 0.2% Triton-X-100% and 0.3 % NP-40] containing Complete protease inhibitor cocktail (Roche, Switzerland), 30 µM MG132 proteasome inhibitor and 30 mM NEM. Lysates from $7 \times 10^6$ cells were prepared as described before and incubated with 3 µg anti-FLAG (Sigma-Aldrich Cat# F3165 RRID:AB_259529), 1:100 anti-Mcl-1 antibody (Abcam Cat# 1239–1 RRID:AB_562163 or Santa Cruz Biotechnology Cat# sc-819 RRID:AB_2144105) or 1:100 anti-myc antibody (Santa Cruz Biotechnology Cat# sc-40 RRID:AB_627268) for 1 hr at 4°C followed by incubation with protein G magnetic beads (Dynabeads, Thermo Fisher Scientific) for 2 hr at 4°C. For co-IP of Cdu1 and Mcl-1, samples were washed briefly and incubated with fresh cell lysate of uninfected cells for 30 min at 4°C. Samples were washed several times and co-precipitates were eluted by addition of 2x SDS-sample buffer and heating to 94°C. For co-IP of Mcl-1 and Bim, samples were washed several times after incubation with protein G beads and directly lysed in 2x SDS-sample buffer. Precipitated proteins were separated by SDS-PAGE and visualized by immunoblotting.

## TUBE pull down

HeLa cells were pre-incubated with 20 µM MG132 4 hr before lysis in TUBE lysis buffer [50 mM Tris-HCl pH 7.5, 150 mM NaCl, 1 mM EDTA, 1 % NP-40 and 10% glycerol] containing Complete protease inhibitor cocktail (Roche), 10 mM N-ethylmaleimide (NEM), 50 µM PR619 DUB inhibitor and 20 µM MG132 proteasome inhibitor. $7 \times 10^6$ cells were lysed for 15 min at 4°C and debris was removed by centrifugation at maximum speed for 5 min. Lysates were incubated with TUBE-agarose for 2 hr at 4°C. Samples were washed with TBST and either eluted by addition of 2x SDS-sample buffer or with 0.2 M glycin pH2.5. Eluted TUBE pull down was analyzed by immunoblot or incubated with 20 nM Cdu1 or 50 nM USP28 in DUB buffer [50 mM Tris HCl pH 7.6, 150 mM NaCl and 2 mM EDTA] for 2 hr at 37°C. The reaction was stopped with 2x SDS-sample buffer and analyzed by immunoblot.

## In vitro binding assay

To proof direct interaction of Cdu1 and Mcl-1, an in vitro binding assay was performed. For the experiment, 5 µg purified recombinant GST-tagged Mcl-1 was bound to glutathione sepharose (GE Healthcare, United Kingdom) in binding buffer [10 mM NaCl, 10 mM NaH₂PO₄, 1 mM EDTA, 0.5% Glycerol, 1% NP-40] and incubated with 5 µg of purified recombinant His-Cdu1 or His-tagged control proteins Cdu2 and a truncated form of chlamydial GroEL for 1 hr at 4°C. After extensive washing, GST-Mcl-1 was eluted by boiling in 2x SDS-sample buffer at 94°C and interaction with Cdu1 or controls was visualized by SDS-PAGE and immunoblot.

## In vitro DUB-assay

The deubiquitinating activity of Cdu1 on poly-ubiquitinated Mcl-1 was demonstrated in an in vitro DUB-assay. To gain poly-ubiquitinated Mcl-1, 2.5 µg purified recombinant GST-Mcl-1 was mixed with 200 ng Ube1 (E1), 150 ng UbcH5c (E2), 500 ng HectH9ΔN (E3) (*Adhikary et al., 2005*) and 50 µg ubiquitin in ubiquitin conjugation initiation buffer (Boston Biochem). The reaction was started by addition of Mg-ATP solution and incubated for 2 hr at 37°C. Equal amounts of poly-ubiquitinated Mcl-1 were incubated with purified recombinant His-Cdu1, His-Cdu1(C345A)-mutant, His-Cdu2, UCH-L3 or by NEM chemically inactivated His-Cdu1 in DUB buffer [50 mM Tris-HCl pH 7.5, 150 mM NaCl, 2 mM EDTA, 2 mM DTT]. After 1 hr incubation at 37°C, the reaction was stopped by mixing the sample with 2x SDS-sample buffer. The ubiquitination status of Mcl-1 was investigated by SDS-PAGE and immunoblot.

## Ubiquitin cleavage assay

20 nM recombinant full length Cdu1 was incubated for 30 min in DUB buffer. 0.5 µg of the different di-ubiquitin chains (Life sensors, Malvern, PA) were added to Cdu1 and incubated for 2 hr at 37°C. The reaction was stopped by addition of 2x SDS-sample buffer and chain cleavage was investigated by an immunoblot assay using an anti-ubiquitin antibody. The amount of Cdu1 was determined using an anti-Cdu1 antibody.

## Whole genome sequencing

Genomic DNA was extracted from renografin purified *C. trachomatis* LGV 434/Bu parental clone, mutant Tn-IGR and mutant Tn-*cdu1* EBs using the Qiagen Blood and Tissue Kit (Qiagen, Germantown, MD). Manufacturer's instructions were used with minor modifications. In brief, 100 µl of purified EBs were aliquoted with 171 µl of Buffer ATL and boiled for 10 min at 96°C. 20 µl of proteinase K was added and the reaction was incubated at 56°C for 1 hr. After incubation, 200 µl buffer AL and 200 µl ethanol (96–100%) was added and mixed thoroughly by vortexing. The reaction was added onto a provided DNeasy Mini spin column and collection tube, centrifuged at 5900 rcf and the flow-through was discarded. Two wash steps were then performed using buffer AW1 and AW2 each with centrifugation and discarding flow-through steps as per the kit's instructions. Finally, 200 µl buffer AE was added to the column and incubated for 20 min at room temperature before centrifugation to obtain eluate with DNA. DNA concentrations were determined by spectrophotometry (OD260/ 280). DNA library preparation (NEBNext Ultra II DNA library kit, Ipswich, MA) and genome sequencing were performed at the genome sequencing core facility at the University of Kansas (https://gsc. ku.edu). The samples were multiplexed on an Illumina Miseq. Paired end (PE100) reads were generated with a Phred score (>Q30) of 95.44%. Sequencing reads were then de-multiplexed and analyzed on the Geneious software suite (version 9.1.7; https://www.geneious.com; RRID:SCR_010519) (*Kearse et al., 2012*). A draft genome was generated for parental, Tn-IGR and Tn-*cdu1* samples and compared to determine genetic differences. After de-multiplexing, each sample had just over 3 million raw reads. The genomes had 98.9 percentage of base pairs covered by more reads than the 10 minimum thresholds. The reads percentages cap out at 98.9 due to repeat regions in the duplicate rRNA loci and variable repeats in the tarp gene. These regions have reads that map to only one copy of the repeat, leading to low coverage in the other repeat, making up 1.1% of the genome. The number of repeats in the tarp gene do not appear to be any different between the Tn-*cdu1* mutant, Tn-IGR and the parental sample. Reference-guided assembly to *Chlamydia trachomatis* 434/ Bu (NCBI NC 010287.1) was performed on parental *C. trachomatis* DNA. Subsequently, the *C. trachomatis* Tn-*cdu1* draft and Tn-IGR draft were generated using the parental sequence as the reference sequence. Alignment files were generated between the *C. trachomatis* parental, Tn-IGR and the Tn-*cdu1* mutant drafts. Plasmid sequences for parental, Tn-IGR and Tn-*cdu1* were assembled using the *C. trachomatis* F-6068 plasmid (CP015307) as a reference. All alignments were performed in Geneious (RRID:SCR_010519) using global alignment tool with free end gaps and a cost matrix of 65% similarity. Mutations that accumulated through the study were annotated and verified using read sequencing data. Mouse fibroblast (L929) DNA contamination was mitigated using the reference-guided assembly techniques. To prevent bias, the transposon insertion site was analyzed both with the reference-guided mapping of the genomes and separately through a reference-guided assembly of the raw reads to the transposon sequence. The read quality and depth of coverage of the transposon assembly was proportional to that of the Tn-*cdu1* and Tn-IGR mutant genomes supporting clonal samples. An assembly of the WT reads to the transposon was performed as a control and no reads were found containing transposon sequence. Genomes for *C. trachomatis* LGV 434/Bu parental (CP019385), Tn-*cdu1* (CP019386), and Tn-IGR (CP019387) were deposited in GenBank.

## Transcervical mouse infections and bacterial burden determination

All animal procedures were performed in accordance with protocols (animal use statement 170–02) approved by the Institutional Animal Care and Use Committee (IACUC) of the University of Kansas. 6–8 week old female C57BL/6 mice (Jackson Laboratories) were treated subcutaneously with 2.5 mg Medroxyprogesteronacetate (Depo-Provera, Pfizer, NY). Seven days later, $5 \times 10^5$ IFU of *C. trachomatis* LGV (434/Bu), *cdu1*::Tn *bla* (Tn-*cdu1*), or IGR (CT383/384)::Tn *bla* (Tn-IGR) were administered to mice transcervically using a non-surgical embryo transfer device (NSET, Lexington, KY). Briefly, NSET devices were inserted into the mouse genital tract allowing direct infection of the upper genital tract (*Gondek et al., 2012*). Five days post-infection, mice were euthanized and uterine horns were extracted and homogenized in 2 ml SPG (Biospec, Bartlesville, OK). Aliquots were frozen at −80°C and DNA isolated using the DNeasy Blood and Tissue Kit (Qiagen, Valencia, CA). Droplet digital PCR (Bio-Rad, Hercules, CA) was used to enumerate *Chlamydia* and host genome copy numbers (*Hindson et al., 2011*; *Roberts et al., 2013*). The following primers and Fam probes were designed for *C. trachomatis secY:* fwd 5' TAA AAA GCC GTG TCA TTC GTC C; rev 5' TCG GCT

TCA ATC ATT GTA CAG C; Probe:/56-FAM/TAATTTACG/ZEN/CTTCCCTTGATCCGGC/3IABkFQ/. The following primers and Hex probes for mouse *rpp30* were designed: fwd 5' CTC TTC CAG TGT GCA AGA AAG C; rev 5' AGT GAC TGA TGA GCT ACG AAG G; Probe:/5HEX/TGAGACGAGTCC TGAGTCTC/3IABkFQ/ (Integrated DNA Technologies, Coralville, IA). PCR reactions were prepared using ddPCR Supermix for Probes and run using Oil-for-Probes droplet emersions using the droplet generator cassette (Bio-Rad). The PCR conditions were: 95°C for 10 min, 40 cycles of 94°C for 30 s, and 98°C for 10 mins followed by cooling to 4°C. Once complete, fluorescent reads of individual droplets were calculated using a QX200 Droplet Reader (Bio-Rad). Data was analyzed using the QuataSoft Software (Bio-Rad) and reported as a ratio of *Chlamydia* DNA to host DNA (*secY/rpp30*, copies/µL). Graphpad Prism 5 (RRID:SCR_002798) was used to generate scatter column chart and perform statistical analysis. One-way ANOVA (Kruskal-Wallis test) with Dunn's multiple-comparison post-test was performed with significance level set to less than 0.01.

## Southern hybridization

For Southern hybridization, long DNA probes were prepared by PCR. Probe 1 was generated using primer pair *bla frw* and *bla rev* resulting in a 1150 bp PCR product amplified from the pGFP::SW2 plasmid. Probe 2 was amplified from chlamydial genomic DNA with primer pair 5' region frw and 5' region rev giving a 1 kb fragment. The PCR products were gel purified and subsequently radiolabeled using the Rediprime II DNA labeling system, GE Healthcare. Chlamydial DNA was prepared form EB stocks using the illustra bacteria genomic Prep Mini Spin Kit (GE Healthcare). The genomic DNA was digested with EcoRI and separated on a 1% agarose gel. The gel was subsequently stained with ethidium bromide and was incubated in 0.125 M HCl for 10 min and then in DNA denaturation buffer (1.5 M NaCl, 0.5 M NaOH) for 30 min. The DNA was transferred to a Nylon-XL membrane (Amersham Hybond-XL, GE Healthcare) by capillary transfer using the denaturation buffer for transfer. After transfer, the membrane was washed with neutralization buffer (3 M NaCl, 0.3 M Tri-sodium citrate, 0.5 M Tris, pH 8.0) for 15 min and was subsequently pre-incubated in hybridization buffer (ULTRAhyb Ultrasensitive Hybridisation Buffer, Ambion, Thermo Fisher Scientific). After 1 hr of pre-incubation, random primed probes were added to the hybridization buffer and incubated overnight at 42°C. Membranes were washed and exposed over night to phosphor storage screens (Fujifilm), which were then scanned by the Typhoon 9200 imager (GE Healthcare). The membrane was stripped with 0.2 M NaOH for 15 min at 42°C every time before a subsequent hybridization with a new probe.

## Cloning, expression and purification of Cdu1 (155-401)

The Cdu1 gene (155-401) was amplified and cloned into a pETM-14 vector by the SLIC method (*Li and Elledge, 2007*). Overexpression was achieved overnight at 16°C in *E. coli* BL21 Star (DE3) cells after induction with 1.0 mM isopropyl *β*-D-1-thiogalactopyranoside at an optical density of 0.8. Purification was pursued through affinity chromatography with Ni-TED beads and cleavage of the poly-histidine tag, followed by size exclusion chromatography utilizing a Superdex 75 16/60 (GE Healthcare) column in 25 mM HEPES (pH 7.5) and 100 mM NaCl. All oligo nucleotides used in this study are listed in *Supplementary file 4*.

## Ub-AMC assay

Cdu1 enzymatic activity was visualized by cleavage of the fluorogenic ubiquitin derivate ub-7-amido-4-methylcoumarin (Ub-AMC) (Boston Biochem). 20 nM recombinant Cdu1 was incubated in DUB-buffer (50 mM Tris HCl pH 7.6, 150 mM NaCl, 2 mM DTT, 2 mM EDTA) for 30 min. As a control, enzymatic activity was blocked by addition of 50 mM N-ethylmaleimide. Pre-incubated enzymes were mixed with 200 nM Ub-AMC and relative fluorescence units (RFUs) were measured in a TECAN multiplate reader (Ex380/Em460 nm).

## Crystallization and X-Ray data collection

As aggregation problems of native Cdu1 (155-401), presumed to be caused by oxidation of surface exposed cysteines, a ΔCys variant of the protein (C174A, C226S, C345A, C386A) was created by site directed mutagenesis. The protein was purified as described above, and crystallization screening was performed by sitting drop vapor diffusion at 20°C. Crystals were obtained at a protein

concentration of 20 mg/ml mixed in a 1:1 ratio with mother liquor containing 100 mM Bicine-NaOH pH 9.0, 10% PEG 20000 and 2% 1, 4 dioxane. Serial seeding using the ΔCys variant crystals was performed with the native Cdu1 construct and crystals were obtained at a protein concentration of 10 mg/ml at 4°C in a solution containing 100 mM Bicine-NaOH pH 9.0, 10% PEG 20000, 2% 1, 4 dioxane and 5% DMSO.

A Cdu1 (155-401) Se-met derivative was prepared by overexpressing the protein in M9-Minimal-Medium supplemented with an amino acid mix containing Selenium-methionine. The protein was purified as described above. Se-Met Cdu1 (155-401) at a protein concentration of 10 mg/ml was supplemented with 6 mM DTT directly prior to crystallization. Crystals were obtained by streak seeding in a precipitant solution containing 100 mM Bicine-NaOH pH 9.0, 10% PEG 20000 and 2% 1,4 dioxane. Crystals were flash frozen in liquid nitrogen with mother liquor containing 100 mM Bicine-NaOH pH 9.0, 10% PEG 20000, 25% PEG 4000 and 2% 1,4 dioxane. Diffraction data were collected at beamlines 14.1 (BESSY, Berlin) and P14 (EMBL/DESY, PETRA III, Hamburg).

### Structure determination and refinement

Data were indexed and integrated with XDS (*Kabsch, 2010*), and further scaling and merging was pursued with the CCP4 suite (*Collaborative Computational Project, 1994*) (RRID:SCR_007255). The structure was solved by the SAD method from peak-data collected from SeMet crystals. The CRANK2 pipeline (*Skubák and Pannu, 2013*) was used for structure solution and initial model building. Refinement was carried out with PHENIX (*Adams et al., 2010*) (RRID:SCR_014224). Data collection and refinement statistics are summarized in *Supplementary file 1*. All figures representing crystal structures were prepared using PyMol (http://www.pymol.org)

## Acknowledgements

We thank Ian Clarke for providing the pGFP::SW2 plasmids, Dirk Bohmann for the HA-ubiquitin construct, Martin Eilers for the ΔN-HECT-H9 construct and the Fbw7 antibody, Roger Davis for the Mcl-1 construct, Theresa Klemm for the recombinant USP28 protein and Prema Subbarayal for the *Ctr* pIncA strain. We thank Alexander Buchberger for valuable advice on the ubiquitination assays and Vera Kozjak-Pavlovic for editing the manuscript. This work was supported by the Deutsche Forschungsgemeinschaft SFB630 project B9 and the priority program 1580 'Intracellular Compartments as Places of Pathogen-Host-Interaction' to TR. KSH, ZED, and PSH and associated research were supported by National Institutes of Health R21AI125929 and P20GM103638. We kindly thank Andreas Schlosser and Stephanie Lamer, from the University of Würzburg for performing the LCMS analysis. In addition, we thank the staff of beamlines BL14.1 of BESSY II in Berlin and P14 of DESY in Hamburg for their excellent support.

## Additional information

### Funding

| Funder | Grant reference number | Author |
|---|---|---|
| National Institutes of Health | R21AI125929 | Kelly S Harrison<br>Zoe Dimond<br>P Scott Hefty |
| National Institutes of Health | P20GM103638 | Kelly S Harrison |
| Deutsche Forschungsgemeinschaft | SFB 630, project B9 | Caroline Kisker<br>Thomas Rudel |
| Deutsche Forschungsgemeinschaft | SPP1580 | Thomas Rudel |

The funders had no role in study design, data collection and interpretation, or the decision to submit the work for publication.

## Author contributions
AF, Data curation, Formal analysis, Investigation, Methodology, Writing—review and editing; KSH, YR, DA, FS, ZD, Investigation; SRC, Data curation, Formal analysis, Investigation, Methodology; BKP, Methodology; CK, Conceptualization, Supervision, Funding acquisition, Project administration, Writing—review and editing; PSH, Conceptualization, Formal analysis, Supervision, Methodology, Writing—review and editing; TR, Conceptualization, Supervision, Funding acquisition, Writing—original draft, Project administration, Writing—review and editing

## Author ORCIDs
Annette Fischer, http://orcid.org/0000-0002-3979-9396
Bhupesh K Prusty, http://orcid.org/0000-0001-7051-4670
Thomas Rudel, http://orcid.org/0000-0003-4740-6991

## Ethics
Animal experimentation: This study was performed in strict accordance with the recommendations in the Guide for the Care and Use of Laboratory Animals of the National Institutes of Health. All animal procedures were performed in accordance with protocols (animal use statement 170-02) approved by the Institutional Animal Care and Use Committee (IACUC) of the University of Kansas.

# Additional files

### Supplementary files
• Supplementary file 1. Data collection and refinement statistics. X-ray crystallographic data collection and refinement statistics for the structure of Cdu1 (155-401). Data for the highest resolution shell are given in parentheses. [a]: Firedel's mates were kept separately for calculation.

• Supplementary file 2. Plasmids. Listed are all plasmids used in this study to transfect human cells or to transform *E. coli* or *C. trachomatis*. If not stated otherwise, constructs were cloned in this study using the oligo nucleotides listed in *Supplementary file 4*.

• Supplementary file 3. *Chlamydia trachomatis* strains. Listed are all *C. trachomatis* strains used and generated in this study. All *C. trachomatis* strains generated by transformation and selection originate from the *C. trachomatis* LGV L2 (434) (ATCC VR-902B) strain.

• Supplementary file 4. Oligo nucleotides. Listed are all oligo nucleotides in $5' \rightarrow 3'$ orientation used in this study. The oligo nucleotides were used for construct cloning, sequencing or southern hybridization as indicated in the comment column.

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
