## [Decision Letter]

Thank you for submitting your article "Chlamydia-containing vacuole serves as deubiquitination platform to stabilize MCl-1 and to interfere with host defense" for consideration by *eLife*. Your article has been reviewed by two peer reviewers, and the evaluation has been overseen by a Reviewing Editor and Ivan Dikic as the Senior Editor. The following individual involved in review of your submission has agreed to reveal his identity: Mads Gyrd-Hansen (Reviewer #2).

The reviewers have discussed the reviews with one another and the Reviewing Editor has drafted this decision to help you prepare a revised submission.

Summary:

This manuscript seeks to address the role that a de-ubiquitinating enzyme (Cdu1) made by the bacterial pathogen *Chlamydia trachomatis* plays in infection of epithelial cells. The premise of the work is that Cdu1 localizes to the surface of the pathogenic vacuole (inclusions) where it recruits the anti-apoptotic factor Mcl1 and stabilizes it by de-ubiquitination, and that this activity contributes to the strong anti-apoptotic state that has been ascribed to cells infected with this pathogen. The mechanism is novel and the premise is appealing. In fact, much of the in vitro binding and deUb activity shown is robust and consistent with what others have reported for ChlaDub, which the authors renamed Cdu1 in this study.

The study addresses an important aspect of infection biology; how pathogens subvert host responses to ensure efficient infection. The study provides convincing evidence that Cdu1 has DUB activity (this is reported previously by other groups) and that Cdu1 inserts into the *Ctr* vacuole. Importantly, the study provides functional evidence that the DUB activity contributes to the infectious capacity of the pathogen in vivo.

Essential revisions:

The main point that needs to be strengthened is the specificity of the role played by Cdu1 in Mcl1 turnover in infected cells including the evidence that Cdu1 exerts its effector function by deubiquitinating and stabilizing MCl-1 and thereby increases cell survival. Overexpression of Cdu1 in *Chlamydia* leading to increased stabilization of Mcl1 in infected cells would be beneficial to demonstrate that Cdu1 plays the key role they propose during infection.

Also better controls are needed and improvement of quantification methods – true source data to review (i.e. raw densitometry reads, including background reads as opposed to the normalized percentages that were the basis for the bar graphs, and quantification of co-localization patterns).

Specific points to be addressed by the authors:

– While ectopic expression of FLAG-Cdu1 (WT and C345A) suggest that Cdu1 can stabilize, MCl-1 levels (Figure 5), the MCl-1 levels were largely similar in *Ctr* Tn-*cdu1* infected cells (cell type is missing in legend) as in WT Ctr-infected cells (Figure 6). Irrespective of this, there also appears to be less mutant Ctr than WT Ctr as judged by the level of cHSP60, which might explain the reduced MCl-1 levels. The biochemical assays to show MCl-1 ubiquitylation are performed by IP of MCl-1 followed by immunoblotting for Ub. The problem with this is that although a "denaturing step" is introduced before the IP, the actual IP is carried out under native conditions and other ubiquitin-modified proteins might therefore be co-purified (specifically or non-specifically). Instead, the authors should purify ubiquitin (by anti-Ub IP or TUBE pulldowns) and immunoblot for MCl-1. Also, ubiquitin IP/pulldown samples should be treated with recombinant DUB (e.g. Cdu1 or another DUB such as USP2 or USP21) to definitely establish that MCl-1 is ubiquitinated.

– The degree of Mcl1 stabilization varies among experiment shown and the timing for analysis chosen for most experiments is odd: the greatest stabilization is seen at 36h, yet most experiments are performed at 20 or 24h, when stabilization is not very impressive. Figure 1—figure supplement 1 at same time point looks very different from what is seen in Figure 1. Figure 1—figure supplement 1 – very minor difference.

The quantification methods used for immunoblots. From the Materials and methods section it appears that they are all performed by chemilumiscence followed by densitometry. Such an approach is highly unreliable without appropriate controls showing that quantification is performed in the linear range of the immunodetection assay.

– Figure 1 shows global down-regulation of TNF-induced Ub of host proteins in *Chlamydia* infected cells. This suggests low specificity of this de-Ub activity and is inconsistent with the premise that Cdu is selective. Are other known targets of Ub (e.g. IKB) similarly affected?

– The experiments showing Cdu1 expression and localization are different in different figures. Figure 4 looks different from Figure 4, and very different from Figure 7—figure supplement 3. In some, it looks like Cdu1 is at the inclusion surface but in other it looks like its on bacteria attached to the inside of the inclusion or within the inclusion- does this localization change over time? Is the FLAG-tagged version behaving like wt? Why is Cdu1 and FLAG not completely co-localized in Figure 3? Figure 7—figure supplement 3 make it look like a lot more Cdu1 is present inside the inclusion (bacterial associated?) by 20h.

– Figure 4 represent the exact experiments (different bacterial markers) but the differences in Mcl1-EGFP recruitment are stark. How representative are these images? What is the quantification? If Cdu is over expressed from a plasmid in *Chlamydia* is more Mcl1 recruited on the inclusion? The staining with Mcl1 antibodies looks strange. It seems like the images were taken on a high focal plane. Quantification?

[Editors' note: further revisions were requested prior to acceptance, as described below.]

Thank you for resubmitting your work entitled "Chlamydia-containing vacuole serves as deubiquitination platform to stabilize MCl-1 and to interfere with host defense" for further consideration at *eLife*. Your revised article has been favorably evaluated by Ivan Dikic (Senior editor), a Reviewing editor and one reviewer.

The manuscript has been improved but there are some remaining issues that need to be addressed before acceptance, as outlined below:

The authors have addressed the major points raised and the new data nicely supports the major conclusions of the study. To avoid confusion, the authors should state in the Introduction that Cdu1 is termed ChlaDUB1 in previous studies (e.g. Misaghi et al. 2006, LeNegrate et al. 2008).

---

## [Author Response]

*Essential revisions:*

The main point that needs to be strengthened is the specificity of the role played by Cdu1 in Mcl1 turnover in infected cells including the evidence that Cdu1 exerts its effector function by deubiquitinating and stabilizing MCl-1 and thereby increases cell survival. Overexpression of Cdu1 in Chlamydia leading to increased stabilization of Mcl1 in infected cells would be beneficial to demonstrate that Cdu1 plays the key role they propose during infection.

*Also better controls are needed and improvement of quantification methods – true source data to review (i.e. raw densitometry reads, including background reads as opposed to the normalized percentages that were the basis for the bar graphs, and quantification of co-localization patterns).*

*Specific points to be addressed by the authors:*

*– While ectopic expression of FLAG-Cdu1 (WT and C345A) suggest that Cdu1 can stabilize, MCl-1 levels (Figure 5), the MCl-1 levels were largely similar in Ctr Tn-cdu1 infected cells (cell type is missing in legend) as in WT Ctr-infected cells (Figure 6). Irrespective of this, there also appears to be less mutant Ctr than WT Ctr as judged by the level of cHSP60, which might explain the reduced MCl-1 levels. The biochemical assays to show MCl-1 ubiquitylation are performed by IP of MCl-1 followed by immunoblotting for Ub. The problem with this is that although a "denaturing step" is introduced before the IP, the actual IP is carried out under native conditions and other ubiquitin-modified proteins might therefore be co-purified (specifically or non-specifically). Instead, the authors should purify ubiquitin (by anti-Ub IP or TUBE pulldowns) and immunoblot for MCl-1. Also, ubiquitin IP/pulldown samples should be treated with recombinant DUB (e.g. Cdu1 or another DUB such as USP2 or USP21) to definitely establish that MCl-1 is ubiquitinated.*

The ectopic overexpression of Cdu1 in HEK cells demonstrates that Cdu1 alone is responsible for the accumulation of MCl-1 in these cells and this is dependent on the enzymatic activity. However, these experiment were performed under overexpression of Cdu1 in the absence of potential regulation by the bacteria. Therefore, we see a strong MCl-1 accumulation and a clear difference between cells expressing either wildtype of mutated Cdu1 enzyme. On the other hand, Chlamydia use several mechanisms to ensure intracellular survival until the developmental cycle is completed. Among them, Chlamydia activate pro-survival signaling pathways which can also result in increased expression of the *MCl-1* gene. It is very likely, that if one pathways contributing to high MCl-1 levels is blocked, for instance Cdu1 dependent stabilization, Chlamydia are able to compensate via other pathways. This might explain why we do not see a strong difference of MCl-1 levels in cells infected with WT or Tn-*cdu1* mutant Chlamydia comparable to the overexpression experiments.

However, a difference in MCl-1 accumulation between Ctr WT and Tn-*cdu1* infected cells is more pronounced if we reduce the intensity of MCl-1 in Figure 6. To confirm these results with equal bacterial load, we repeated the infection experiments with Ctr WT and Tn-*cdu1* and performed the readout for MCl-1 stabilization at 24 and 30 hpi. The quantification revealed a clear difference in MCl-1 stabilization in Wt and mutant-infected cells (see new Figure 6 and Figure 6—figure supplement 1). The results show that MCl-1 stabilization increases with infection progression.

As suggested by the reviewers, we performed TUBE pull down of ubiquitinated proteins and probed the pull down for MCl-1 in immunoblot. As already shown in the denaturing IP of MCl-1, there is less ubiquitinated MCl-1 in Ctr WT-infected cells compared to Ctr Tn-*cdu1*-infected HeLa cells (see Figure 7—figure supplement 1). As suggested by the reviewers, we probed the TUBE-PD for p53, another known ubiquitin-target. However, we do not see any difference in p53-ubiquitination in cells infected with Ctr WT or Tn-*cdu1* in TUBE-PD samples.

Furthermore, we isolated ubiquitinated proteins by TUBE pull down out of uninfected HeLa cells and incubated the eluted proteins with recombinant 20 nM Cdu1 or 50 nM USP28 protein. Subsequently, the reduction on MCl-1 ubiquitination was analyzed by immunoblot. As seen in Figure 5—figure supplement 1, only incubation with recombinant Cdu1 but not with USP28 leads to deubiquitination of MCl-1.

We have described the TUBE pull down and treatment with DUBs in the Materials and method section.

– The degree of Mcl1 stabilization varies among experiment shown and the timing for analysis chosen for most experiments is odd: the greatest stabilization is seen at 36h, yet most experiments are performed at 20 or 24h, when stabilization is not very impressive. Figure 1—figure supplement 1 at same time point looks very different from what is seen in Figure 1. Figure 1—figure supplement 1 – very minor difference.

*The quantification methods used for immunoblots. From the Materials and methods section it appears that they are all performed by chemilumiscence followed by densitometry. Such an approach is highly unreliable without appropriate controls showing that quantification is performed in the linear range of the immunodetection assay.*

We apologize for the poor description of the experiment in the figure legend for Figure 1—figure supplement 1. The infected cells were treated with CHX 24/26/28 h after infection and all samples were lysed at 30 hpi resulting in the different duration of CHX treatment but the same infection time. Therefore, the amount of MCl-1 seen in the untreated sample resembles 30 hpi and not 24 hpi. Furthermore, minor variations in MCl-1 can result from differences in the basal MCl-1 level of the cells.

As already indicated above, we analyzed MCl-1 stabilization at 24 and 30 hpi in Ctr-infected cells (Figure 6). And in accordance with the reviewers’ statement, there is more MCl-1 stabilization at later time points of infection.

As suggested by the reviewers in one of the following points, we also performed timeframe experiments to monitor Cdu1 secretion by the different *Chlamydia* strains (Figure 3—figure supplement 2, Figure 3—figure supplement 4 and Figure 6—figure supplement 1). As seen in the pictures, upon 30 or 36 hours of *Chlamydia* infection the inclusion expands to fill large parts of the cell.

Performing immunofluorescence experiments to analyze protein recruitment at 30 or 36 hpi would possibly give false positive results. Furthermore, during the late phase of infection, *Chlamydia* secrete multiple proteases into the inclusion lumen which make experiments like immunoprecipitation or TUBE pull down experiments difficult due to protein degradation. We therefore chose the 24 hpi time point as suitable for all experiments (MCl-1 expression, immunofluorescence, IP). However, the TUBE-pull down experiments were now performed at 27 hpi.

Other as suspected by the reviewers, the quantification of the immunoblots was not performed by densitometry but by quantitative chemiluminescence with the help of an INTAS imaging system as it was stated in the Materials and methods section.

*– Figure 1 shows global down-regulation of TNF-induced Ub of host proteins in Chlamydia infected cells. This suggests low specificity of this de-Ub activity and is inconsistent with the premise that Cdu is selective. Are other known targets of Ub (e.g. IKB) similarly affected?*

We don’t see strong global down-regulation of protein ubiquitination upon *Chlamydia* infection and TNFα treatment (see quantification in Figure 1). We included the TNFα treatment in this experiment to induce MCl-1 ubiquitination by the cell. However, we do not see any difference in MCl-1 ubiquitination between uninduced and TNFα-induced samples but there is less MCl-1 ubiquitinated in *Chlamydia*-infected cells independent of TNFα-stimuli which is the major finding of this experiment. Furthermore, as depicted in Figure 7, there is no difference in the global amount of ubiquitinated proteins in uninfected and Chlamydia WT- as well as Tn-*cdu1*-infected cells.

In addition, an overexpression of Cdu1 in cells (see Figure 5) does not reduce the global amount of ubiquitinated proteins which strengthens the hypothesis of Cdu1 being selective. Till now, we did not identify all substrates of Cdu1, but we believe that besides MCl-1 there are additional proteins targeted by Cdu1.

Moreover, the results from Figure 1 were obtained from *Chlamydia*-infected cells and even if there is a minor global downregulation of protein ubiquitination after TNFα-treatment in infected cells it could still be the result of any other chlamydial strategy to manipulate the host cell. It is also known that *Chlamydia* manipulate multiple signaling pathways which are often dependent on protein ubiquitination or deubiquitination and just by blocking or activating these pathways (independent of Cdu1-activity) small changes in protein ubiquitination would be detectable.

To address the specificity of Cdu1, we included p53 as a high turnover protein which is involved in apoptosis signaling as an additional control in the TUBE experiments. The results are depicted in Figure 5—figure supplement 1 and Figure 7—figure supplement 1.

*– The experiments showing Cdu1 expression and localization are different in different Figures. Figure 4 looks different from Figure 4, and very different from Figure 7—figure supplement 3. In some, it looks like Cdu1 is at the inclusion surface but in other it looks like its on bacteria attached to the inside of the inclusion or within the inclusion- does this localization change over time? Is the FLAG-tagged version behaving like wt? Why is Cdu1 and FLAG not completely co-localized in Figure 3? Figure 7—figure supplement 3 make it look like a lot more Cdu1 is present inside the inclusion (bacterial associated?) by 20h.*

To monitor the dynamics of Cdu1 secretion, we performed timeframe experiments and detected *Chlamydia* and Cdu1 by immunofluorescence microscopy. Upon 20 h of infection, Cdu1 starts to be secreted to the inclusion surface but is also present within the bacterial particles (Figure 3—figure supplement 4 and Figure 6—figure supplement 1). Already 24 hpi, major amounts of Cdu1 are present on the inclusion surface. At 30 hpi, there is even more Cdu1 secreted, but almost the whole cytoplasm of the cell is now filled up by the inclusion (see the nucleus which is visible as a background staining of the Cdu1 antibody). Working with cells infected for 30 h is quite challenging and proper analysis of MCl-1 recruitment to the inclusion by immunofluorescence staining is almost impossible. Therefore, we chose the time point of 24 to 26 hpi for all performed experiments to be able to compare the results of the different experiments.

We now changed the infection protocol to a reduced volume of infection media and a media exchange 1 hpi to reduce the variations in developmental cycle progression.

*– Figure 4 represent the exact experiments (different bacterial markers) but the differences in Mcl1-EGFP recruitment are stark. How representative are these images? What is the quantification? If Cdu is over expressed from a plasmid in Chlamydia is more Mcl1 recruited on the inclusion? The staining with Mcl1 antibodies looks strange. It seems like the images were taken on a high focal plane. Quantification?*

The GFP-expression in the two representative cells is very different and therefore the GFP-Mcl1 recruitment in the background of the strong expression is not that visible. We exchanged the representative pictures with equal GFP-protein expression. See new Figure 4.

The transfection rate to obtain GFP-MCl-1 positive was relatively low. Therefore, we quantified MCl-1-decorated inclusions from fixed cells stained for MCl-1, cHSP60 (*Chlamydia*) and DAPI and analyzed them by confocal microscopy (see representative pictures in Figure 4—figure supplement 1, Figure 6—figure supplement 2+B). The quantification of rings was done using the cell counter plugin (FIJI) followed by automated counting of the cell nucleus (stained with DAPI) and the number of chlamydial inclusions (stained against HSP60) using the object count plugin form (FIJI). The percentage of chlamydial inclusions decorated with rings was calculated by diving the number of rings by the total number of chlamydial inclusions observed in each picture. Details are described in the Materials and methods section starting in subsection “Immunolourescence microscopy and image processing”. Figure 6 shows the relative number of MCl-1-decorated chlamydial inclusions of WT and Tn-*cdu1*-infected cells with a significant lower number in case of mutant inclusions.

To address, if inactivation of Cdu1 reduces or Cdu1 overexpression increases amounts of MCl-1, we analyzed area and intensity of MCl-1 rings around the chlamydial inclusions. For comparison of Ctr WT vs. Tn-*cdu1* the above described pictures were analyzed. In addition, HeLa cells infected with Ctr pTet/Cdu1-FLAG and induced for Cdu1 overexpression by addition of AHT were fixed 24 hpi and stained for MCl-1, cHSP60 and DAPI. Intensity of the MCl-1 rings decorating chlamydial inclusions in WT, Tn-*Cdu1*, pTET Cdu1 *Chlamydia* (with and without AHT treatment) were calculated using FIJI after appropriate thresholding for background. Samples were measure for area and intensity of the MCl-1 ring. The absolute intensity was calculated by dividing the observed arbitrary intensity value by the area of the ring which was then plotted to produce the graphs. Results of MCl-1 ring intensity upon Cdu1 inactivation (WT vs. Tn-*cdu1*) or overexpression (pTet/Cdu1 -/+ AHT) are depicted in Figure 6 and Figure 6—figure supplement 2, respectively. The quantification of these experiments showed a correlation of Cdu1 quantities and MCl-1 stabilization.

During confocal microscopy of *Chlamydia*-infected cells we set the focal plane to the inclusion and not the nucleus or basal part of the cell.